# On more accurate alignment modeling methods for automatic speech recognition

## Abstract

The connectionist temporal classification (CTC) training criterion optimizes the conditional log probability of the label sequence given the input, which involves a sum over all possible alignment label sequences including blank. It is well known that CTC training leads to peaky behavior where blank is predicted in most frames and the labels are focused mostly on single frames. Thus, CTC is suboptimal to obtain accurate word boundaries. Hidden Markov models (HMMs) can be seen as a generalization of CTC and trained in the same way with a generalized training criterion, and may lead to similar problems. Label units such as subword units and its vocabulary size or phoneme-based units also significantly impact the alignment quality. Here we study different methods of obtaining an alignment with the goals to improve alignment quality while keeping a good performing model, and to gain better understanding of the training dynamics. We introduce (1) a synthetic framework to study alignment behavior, and compare various models, noise and training conditions, (2) a new training variant with renormalizing the gradients to counteract the class imbalance of blank, (3) a novel CTC model variation to use a hierarchical softmax and separating the blank label in CTC, as another alternative to counteract class imbalance, (4) a novel way to get alignments via the gradients of the label log probabilities w.r.t. the input features. This method can be used for all kinds of models, and we evaluate it for CTC and attention-based encoder-decoder (AED) subword based models where it performs competitive and more robustly, although phoneme-based HMMs still provide the best alignments.

## 1 Introduction

Current sequence-to-sequence models (Prabhavalkar et al., 2023) such as connectionist temporal classification (CTC) (Graves et al., 2006) can be trained from-scratch using the sequence-level cross-entropy and summing over all alignments. However, CTC alignments tend to be dominated by blanks, causing a peaky behavior (Zeyer et al., 2021; Huang et al., 2024), which can be suboptimal to obtain good alignments.

CTC can be seen as a special case in the broader hidden Markov model (HMM) framework with a simplified label topology including the blank label instead of silence, without label priors and without explicit transition probabilities (Zeyer et al., 2017; Hadian et al., 2018; Raissi et al., 2022; Zhao & Bell, 2022). When hybrid neural network (NN)-HMMs are trained with the sum over all alignment paths, when no prior and no transition probabilities are used, the same peaky behavior occurs (Zeyer et al., 2017; 2021). Previous work addressed the issue of peaky behavior by using priors during training (Zeyer et al., 2021; Chen et al., 2023; Huang et al., 2024). When a prior is used, it can also switch to the opposite extreme with no silence at all and very bad alignments (Zeyer et al., 2017; Raissi et al., 2022).

It is known that the Gaussian mixture hidden Markov model (GM-HMM) alignments offer more reliable segment and word boundaries. An input feature representation that is obtained from current neural architectures that use self-attention or recurrent layers is substantially different from the one used in GMM. The neural encoder has the freedom to displace the output label with respect to their ground truth position. It can shift and compress parts of the signal, or even reverse the signal in the time dimension (Schmitt et al., 2024). This expressive capacity ultimately also allows for peaky

behavior and otherwise potential bad alignment quality. Thus, training dynamics and the training criterion play an important role for the alignment behavior and quality.

Here we study different variations of the training criterion and training procedure with the goals:

- to improve alignment quality,
- to improve convergence rate and training robustness,
- to improve the recognition performance, and
- to gain better understanding of the training dynamics.

We compare the alignment quality in terms of time stamp error (TSE), which is the average absolute distance of word left/right boundaries and word center positions, compared to a GMM alignment as reference (Zhang et al., 2021; Raissi et al., 2022). We also measure the amount of silence (or blank) in the alignment, where a high amount of silence indicates more peakiness. The model performance is evaluated by the word error rate (WER).

Our contributions in this work are:

- A framework to study alignment behavior based on artificially generated data, and compare various model, noise and training conditions.
- A new training variant: normalized gradients as an alternative to training with prior.
- A novel CTC model variation: Separating the blank label in CTC, as another alternative to counteract class imbalance.
- A novel way to get alignments via the gradients of the label log probabilities w.r.t. the input features, leading to higher alignment quality.

In terms of WER, we find only small improvements using the new training variant or model variant.

Note, there is a wide range of related works (see Appendix A.1). In many cases, when the alignment quality is very good (e.g. using a prior as Huang et al. (2024), or GMMs), the model is bad in terms of WER performance. Here we start with our best CTC models (in terms of WER) as baseline, and try to extract good alignments from them. We want a model which is both good in terms of WER and can generate a good alignment.

## 2 MODELS & TRAINING CRITERIA

Let $x_1^{T'}$ be the input sequence of length $T'$, e.g. some log mel or Gammatone features. We use a downsampling convolutional frontend with $T = \lceil T'/F \rceil$ for $F = 4$ or $F = 6$ together with a Conformer encoder (Gulati et al., 2020):

$$x'^{T}_1 = \text{Frontend}(x_1^{T'}) \tag{1}$$
$$h_1^T = \text{Encoder}(x'^{T}_1) \tag{2}$$

Let $a_1^S$ be the output sequence of labels of length $S$ with $a_s \in \mathcal{A}$. Let $y_1^T$ be the alignment label sequence over the time frames with $y_t \in \mathcal{Y}$. In case of CTC, we use $\mathcal{Y} = \mathcal{A} \cup \{\epsilon\}$, i.e. $y$ is either one normal label ($\mathcal{A}$) or otherwise the special *blank* symbol $\epsilon$.

In case of CTC and HMM, then we define the (unnormalized) logits for the alignment labels $\mathcal{Y}$ in time frame $t$ together with the alignment label probability distribution as:

$$z_t = \text{Linear}(h_t) \in \mathbb{R}^{\mathcal{Y}} \tag{3}$$
$$p(y_t{=}y \mid h_t) = \text{softmax}_{\mathcal{Y}}(z_t)_y \tag{4}$$

**CTC**

$$L_{\text{CTC}} = -\log \sum_{y_1^T : a_1^S} \prod_t p(y_t \mid h_t) \tag{5}$$

**HMM** The (hybrid) HMM can be seen as a generalization of CTC in that various label topologies are possible, i.e. the mapping of $\mathcal{A}$ to $\mathcal{Y}$ and what type of alignment labels $\mathcal{Y}$ are used. Usually, there is no blank but a silence label instead, which is not allowed within words.

$$L_{\text{HMM}} = -\log \sum_{(y_1^T, s_1^T)\,:\, a_1^S} \prod_t \frac{p(y_t \mid h_t)^\alpha}{p(y_t)^\beta} \cdot p(s_t \mid s_{t-1})^\gamma \tag{6}$$

Note that we have scales $\alpha$, $\beta$ and $\gamma$ here for the posterior, prior and transition models respectively. When putting $\alpha = 1, \beta = 0, \gamma = 0$, and when using the CTC label topology with blank, we see that CTC is a special case of the HMM training criterion.

The prior model $p(y)$ can be estimated given a reference alignment or the transcriptions. In alternative, it can also be average of the posterior over time frames for a given utterance or even on the whole training data. The prior and/or the transition model both significantly impact the alignment behavior and accuracy.

**AED** This model directly defines $p(a_s \mid a_1^{s-1}, h_1^T)$, which uses the cross-attention mechanism to attend to $h_1^T$, and then finishes with an end-of-sequence (EOS) label at the end Chorowski et al. (2015); Chan et al. (2016). There are no explicit alignments in this model (no alignment labels $y$). The loss is defined as:

$$L_{\text{AED}} = -\log \prod_s p(a_s \mid a_1^{s-1}, h_1^T) \tag{7}$$

## 3 Training with Normalized Gradients

The use of the prior in the HMM training criterion can also be interpreted as a way to rebalance the loss with the inverse frequencies of the alignment classes. Specifically, blank or silence will be the most common label (even when not peaky). Thus this will dominate in the training criterion and in its gradients, and the prior rebalances this.

We studied the gradients of the normal CTC training criterion (when there is no prior used) and how to modify (weight) the gradients such that the loss gradient influence is totally balanced across the label classes. For CTC, the gradient of the loss w.r.t. the logits is

$$\nabla_{z_{t,j}} L_{\text{CTC}} = p(y{=}j \mid h_t) - \upsilon_{t,j} \tag{8}$$

where

$$\upsilon_{t,j} = \frac{\sum_{y_1^T\,:\, a_1^S, y_t = j} \prod_t p(y_t \mid h_t)}{\sum_{y_1^T\,:\, a_1^S} \prod_t p(y_t \mid h_t)} \tag{9}$$

is the soft-alignment[1]. The soft-alignment $\upsilon_t$ is a frame-wise probability distribution over the alignment labels $\mathcal{Y}$, i.e. $\sum_j \upsilon_{t,j} = 1$. The soft-alignment $\upsilon_{t,y}$ is the target for $p(y \mid h_t)$ in training (the optimum is reached when $p(y \mid h_t) = \upsilon_{t,y}$). Thus, the inverse of the expected value of the soft-alignment[2]

$$\overline{\upsilon} = \mathbb{E}_t \upsilon_t \in \mathbb{R}^{\mathcal{Y}} \tag{10}$$

can be used to rescale the loss. In practice, we only modify the gradient here and not the loss itself. Specifically, we use

$$\nabla_{z_{t,j}} L_{\text{NormedGradCTC}} = \nabla_{z_{t,j}} L_{\text{CTC}} \cdot \min\left(\max\left((\overline{\upsilon} \cdot |\mathcal{Y}|)^{-1}, \overline{\upsilon}_{\min}\right), \overline{\upsilon}_{\max}\right). \tag{11}$$

---

[1] Also called Baum-Welch alignment. This can be computed via the forward-backward algorithm, i.e. using dynamic programming. Or this can be computed implicitly using the forward algorithm and automatic differentiation.

[2] It can be calculated over the time frames $t$ of the current sequence, or also the current mini-batch. We found that the mini-batch works a bit better.

The factor $|\mathcal{Y}|$ scales $\overline{\upsilon}$ back to its original range[3], and the clamping is added to make it more robust against outliers[4]. Note, this is now a scaling per vocab. dimension in $\mathcal{Y}$, unlike some other methods which would perform the scaling per time frame. However, for framewise CE training, where such scaling by prior is sometimes used, these are equivalent.

This is very related to the training criterion with a prior: Instead of the prior (which is e.g. estimated on the average of $p(y \mid h)$), now we use the prior estimated on the average of the soft alignment $\upsilon$.

Consider also the case of framewise CE training

$$L_{\text{framewise}} = -\sum_t \log p(\overline{y}_t \mid h_t) \tag{12}$$

for a given alignment $\overline{y}$. We get the gradient

$$\nabla_{z_{t,j}} L_{\text{framewise}} = p(y{=}j \mid h_t) - \upsilon'_{t,j} \tag{13}$$

$$\upsilon'_{t,j} = \delta_{j=\overline{y}_t}, \tag{14}$$

and $\overline{\upsilon'}$ when estimated on the whole training data becomes the classical count-based prior.

Consider also the case of very clean synthetic data together with a simple single-layer feed-forward neural network (FFNN) (see Section 6.1), where we can initialize $W = \mathbb{1}$ and $b = 0$. This initialization will provide a perfect alignment for this synthetic task. It will stay perfect as long as $b$ stays uniform. Now, $\nabla_b L_{\text{CTC}}$ is not uniform, thus the model will not keep good alignment behavior. But $\nabla_b L_{\text{NormedGradCTC}}$ is uniform by construction. When using CTC with prior, $\nabla_b L$ would also not be uniform, i.e. $L_{\text{NormedGradCTC}}$ is really the best possible loss you can have here.

# 4    SEPARATION OF THE BLANK LABEL IN CTC

In this modeling approach, we use a separate sigmoid unit for the blank label $\epsilon$, and the softmax over all remaining non-blank labels $\mathcal{A}$. Specifically:

$$p'_{\mathcal{Y}}(y_t{=}y \mid h_t) = \begin{cases} p_\epsilon(\epsilon \mid h_t), & y = \epsilon \\ (1 - p_\epsilon(\epsilon \mid h_t)) \cdot p_{\mathcal{A}}(y \mid h_t), & y \in \mathcal{A} \end{cases} \tag{15}$$

$$p_\epsilon(\epsilon \mid h_t) = \sigma(z_{t,\epsilon}) \tag{16}$$

$$p_{\mathcal{A}}(y \mid h_t) = \text{softmax}_{\mathcal{A}}(z_{t,\mathcal{A}})_y \tag{17}$$

where $\sigma(z) = \frac{1}{1+\exp(-z)}$ is the sigmoid function.

This has been used before for transducer models (Variani et al., 2020; Zeyer et al., 2020), however, it has never been used for CTC. This is like a hierarchical softmax (Morin & Bengio, 2005) where the first decision is between blank and all other labels.

Consider the case of framewise CE training with a reference alignment $\overline{y}_1^T$, i.e. the loss

$$L_{\text{framewise}} = -\sum_t \log p(\overline{y}_t \mid h_t). \tag{18}$$

In this case, the classes in $p_{\mathcal{A}}$ are much more balanced compared to the classes in $p_{\mathcal{Y}}$, as blank is usually the most imbalanced class.

Independent of the separation, for the gradient of the sequence loss $L_{\text{CTC}}$ w.r.t. the logits $z$, we get

$$\nabla_{z_{t,j}} L_{\text{CTC}} = -\nabla_{z_{t,j}} \sum_i \text{stopgrad}(\upsilon_i) \cdot \log p(y{=}i \mid h_t) \tag{19}$$

where $\upsilon$ is the soft alignment (see Equation (9)).

For the full softmax (non-separated blank), we get

$$\nabla z_j \sum_i \text{stopgrad}(\upsilon_i) \cdot \log p_{\mathcal{Y}}(y{=}i \mid h_t) = \upsilon_j - \text{softmax}_{\mathcal{Y}}(z)_j. \tag{20}$$

---

[3]Consider $\overline{\upsilon} = \frac{1}{|\mathcal{Y}|}$ when uniform.
[4]We use $\overline{\upsilon}_{\min} = 0.5$, $\overline{\upsilon}_{\max} = 1.1$.

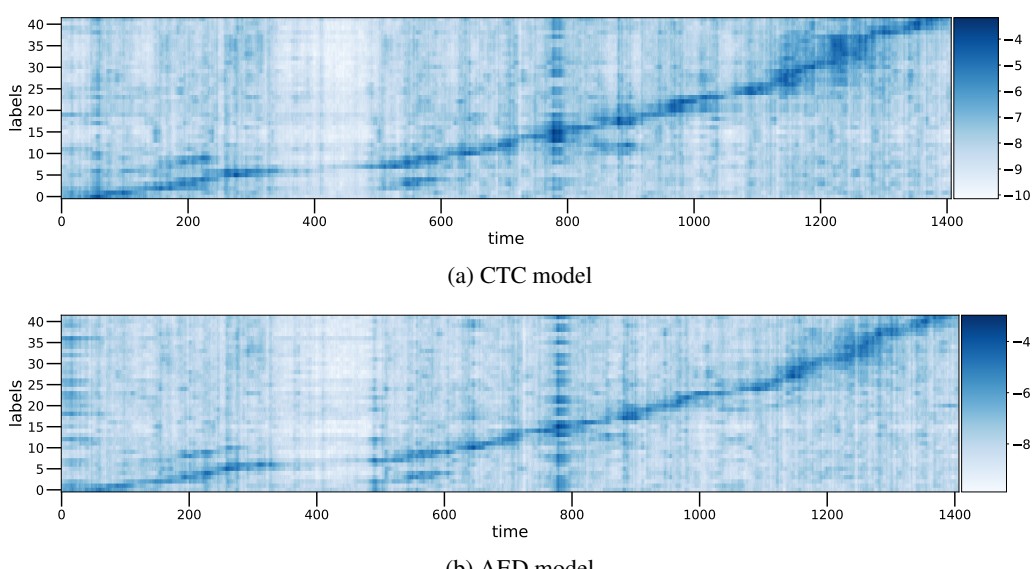

(a) CTC model

(b) AED model

Figure 1: The gradient norms used for alignment generation, specifically $\log \operatorname{softmax}_{\bar{t}}(G) \in \mathbb{R}^{T \times S}$, i.e. the log softmax of log norm of $p(a_s \mid a_1^{s-1}, x_1^{T'})$ w.r.t. the input frames $x_1^T$. Sequence train-clean-100/103-1240-0000.

With separated blank, we get[5]:

$$\nabla z_j \sum_i \operatorname{stopgrad}(v_i) \log p'_{\mathcal{Y}}(y{=}i \mid h_t) = \begin{cases} v_\epsilon - \sigma(z_{t,\epsilon}), & j = \epsilon, \\ \left( \frac{v_j}{1-v_\epsilon} - \operatorname{softmax}_{\mathcal{A}}(z_{t,\mathcal{A}})_j \right) \cdot (1 - v_\epsilon), & j \neq \epsilon \end{cases} \tag{21}$$

We can see that $\frac{v_j}{1-v_\epsilon}$ becomes the soft target for $p_{\mathcal{A}}(y \mid h_t)$, and this loss is scaled by $1 - v_\epsilon$.

This also allows for faster greedy decoding and faster framewise CE training (given a fixed reference alignment). See Appendix A.7 and Appendix Table 15.

## 5 ALIGNMENTS VIA GRADIENTS

The encoder is often so powerful that it can shift around the signal (e.g. with streaming models) or even reverse the time dimension (Schmitt et al., 2024)[6]. When this happens, the normal forced align-ment quality degrades, while the gradient-based alignment always gives a meaningful alignment, as this uses the gradient w.r.t. the input signal.

We can calculate the gradient of the log probability of some target label given some input frame $p(a_s \mid a_1^{s-1}, x_1^{T'})$ w.r.t. the input frame $x_t$. Comparing the norm of these gradients over the time frames $t$ will give us some indication on the importance of each frame for this specific output label $a_s$. Specifically, we calculate the log norm[7]

$$G_{s,t} := \log \left\| \nabla_{x_t} \log p(\bar{a}_s \mid \bar{a}_1^{s-1}, x_1^{T'}) \right\|_p \in \mathbb{R}. \tag{22}$$

An example of the matrix $\log \operatorname{softmax}_{\bar{t}} G$ can be seen in Figure 1a and Figure 1b. The alignment can clearly be seen in this matrix.

Note that this is straightforward to compute for an AED model (we exclude the EOS label here), and was done in a similar way in Schmitt et al. (2024). It is possible for a CTC model or HMM as well,

---

[5]For the full derivation, see Appendix A.2.

[6]Arguably reversing the time dimension will not happen for CTC, though.

[7]We found that the log norm was better scaled than the norm, and yielded better results. We also tested different $p$-norms, and found $p = 0.1$ in most cases to perform best.

using the prefix scores (Hori et al., 2017)

$$\log p_{\text{CTC}}(\overline{a}_s \mid \overline{a}_1^{s-1}, x_1^{T'}) = \log \sum_{t \leq T, y_1^t \,:\, a_1^s} p(y_1^t \mid h_1^t) - \log \sum_{t \leq T, y_1^t \,:\, a_1^{s-1}} p(y_1^t \mid h_1^t). \quad (23)$$

which can be calculated efficiently using dynamic programming[8]. As a further tweak, we slightly modify the gradients of the logits by masking out the gradients of the blank logit. I.e. in the automatic differentiation, we hook after the gradient computation of $\nabla_z L$, and then $(\nabla_z L)_\epsilon \leftarrow 0$. This slightly improves our results (see Appendix A.8).

To use this to get some alignment, we need to define what kind of alignment label topology we allow (mapping $a_1^S$ to $y_1^T$) and how to score one particular alignment $y_1^T$ such that we can search for the one with the highest score.

For the label topology, we use $\mathcal{Y} = \mathcal{A} \cup \{\epsilon\}$ (like CTC). We allow any number of $\epsilon$ (blank) labels between any of the real labels, we allow the real label to be repeated multiple times over the time frames $t$. This is very similar to the CTC label topology except that we do not enforce an $\epsilon$ between two equal labels (when $a_s = a_{s+1}$). This can be formulated as a finite state automaton with enumerated states $Y_1^{2S+1} = (\epsilon, 1, \epsilon, 2, \ldots, S, \epsilon)$. We search for an allowed state sequences $r_1^T : a_1^S$ for state indices $r_t \in \{1, \ldots, 2 \cdot S + 1\}$ which maximizes

$$\text{GradScore}(r_1^T) = \sum_{t=1}^{T} \text{GradScore}(r_t) \quad (24)$$

$$\text{GradScore}(r_t) = \begin{cases} \log \text{softmax}_{\overline{t}}(G)_{Y_{r_t}, t}, & Y_{r_t} \neq \epsilon, \\ \gamma_\epsilon, & Y_{r_t} = \epsilon \end{cases} \quad (25)$$

for some fixed blank score $\gamma_\epsilon$ hyperparameter (usually $\gamma_\epsilon = -6$). The best $r_1^T$ can be found via dynamic programming. We obtain the final alignment label sequence $y_1^T$ with $y_t = \begin{cases} a_{Y_{r_t}}, & Y_{r_t} \neq \epsilon, \\ \epsilon, & Y_{r_t} = \epsilon \end{cases}$.

We experimented with variations of GradScore and came up with GradScoreExt where we use a better estimate of the blank score and then also renormalize over the labels including blank. The exact definition is in appendix Equation (49).

# 6 EXPERIMENTAL SETUP

Both the phoneme-based models and subword-based models use a Conformer encoder (Gulati et al., 2020). See Appendix A.4 for further details. All the code for all the experiments will be published.

## 6.1 SYNTHETIC DATA

We can create synthetic data and simulate the speech recognition task to various degrees of complexity and difficulty. This allows to study the alignment behavior under very controlled conditions. The data synthesis starts by sampling a ground truth reference alignment from a given probability distribution, and then creates corresponding input features very directly from the alignment, such that the model just needs to learn an identity function. When the data is designed to be as simple and clean as possible, and by design unambiguous, which variant of training criterion and modeling converges to the ground truth alignment? The training criterion does not have any explicit aspect about the alignment, and there are many global optima which would yield a very bad alignment. Specifically, we design the framework such that we control:

- The ground truth alignment. We construct the input features accordingly.
- Noise in the input features.
- The vocabulary and labels, and statistics on how many words per sequence.
- Statistics about how much silence there is and the duration of labels. This indirectly simulates different framerates of the input features.

---

[8]In fact, calculating the prefix scores is already part of the usual CTC loss calculation itself.

Then we support a variety of model types (GMM, CTC, hybrid HMM; various neural encoders; different prior model variants; transition probabilities), CTC and HMM label topology, and different training criteria.

See Appendix A.4.5 for a detailed description of how we sample an input-target pair from the dataset. An input feature $x$ is just a one-hot encoding of the corresponding target label $y$. From this construction, there is a trivial optimal mapping from the (non-noisy) input features to the target probability distribution $p(y \mid x)$:

$$p_{\text{opt}}(y{=}i \mid x_t) = x_{t,i} \tag{26}$$

Now, when we use a simple single layer feed-forward neural network (FFNN), i.e. the model

$$p_{\text{FFNN}}(y \mid x) = \text{softmax}_{\mathcal{Y}}(x \cdot W + b), \tag{27}$$

with $W \in \mathbb{R}^{D \times \mathcal{Y}}, b \in \mathbb{R}^{\mathcal{Y}}$, we reach a similar optimal solution as close as we want with the scaled identity matrix $W = \mathbb{1} \cdot c$ for some large constant $c$ and $b = 0$.

# 7 EXPERIMENTAL RESULTS

## 7.1 PRIORS AND TRANSITIONS FOR HMM AND CTC

### 7.1.1 EXPERIMENTS ON SYNTHETIC DATA

We compare different types of prior probabilities using a simple feed-forward neural network (FFNN) (Appendix Table 7). The static prior (using the real ground truth) interestingly performs bad, even after tuning the scales. The average of the posterior model with stop gradient works best. This is the only prior type which really works here. We also see that the scales are important here. It works without them but it is slightly suboptimal. No prior also has problems here. Normally, no prior would work and result in peaky behavior, but this is not really possible with the FFNN here, and also the HMM label topology is suboptimal for that.

We compare different dataset distributions (Appendix Table 10). Note that the number of frames per label relates to the framerate on real data. For Switchboard, the average length of a phoneme is 80ms. When the model operates on a 40ms framerate, that corresponds to about 2 frames per phoneme label. We see that the convergence problems mostly occur only with a high number of frames per label, i.e. with a high frame rate (see Appendix Table 10). Specifically, for the high framerate ($N_{\text{word}} = 10$), using prior together with posterior is important to get good results, and using posterior alone does not work at all[9] while for low framerate ($N_{\text{word}} = 2$), prior together with posterior still works, but is slightly suboptimal, and using the posterior alone reaches the optimal result.

Here we are using more realistic settings: Using noise, a more powerful posterior BLSTM model (Schuster & Paliwal, 1997; Hochreiter & Schmidhuber, 1997), HMM label topology, a higher batch size and a more realistic dataset distribution. Results are in Table 1. Using a too high posterior scale breaks it, but otherwise, it usually works. There are configurations where only the transition model is helpful, and same with only prior, although only transition model seems better. The best result is achieved with using both the prior and the transition model. Note that we are never able to achieve zero LER or zero TSE here. The amount of noise might be unrealistically high now.

### 7.1.2 PHONEME-BASED MODELS

The results presented in this section for the zero order label context phoneme based models using real data (LibriSpeech and Switchboard) show the effect of transition probability and prior for the HMM based systems.

The experiments that are conducted for phoneme based models share the same experimental setups for both HMM and CTC. However, they are not directly comparable to the experiments on CTC presented in following sections. Here, we use fewer epochs on LibriSpeech (25 instead of 100) and we use a different software framework. We show the effect of the reduction of number of epochs in Appendix Table 11.

---

[9]This is still a FFNN; it does work with more powerful models.

Table 1: Comparing the **posterior/prior/transition scales** with HMM label topology in the presence of noise ($\sigma_\xi = 0.5$) on **synthetic data** with a 2-layer BLSTM posterior model and higher batch size 100. Prior is via posterior average with stop gradient. The experiment is repeated over 10 random seeds to measure the mean $\mu$ and standard deviation $\sigma$. The label-error-rate (LER) provides an indicator of the performance of the model. We calc. framewise (fw.) CE using the reference alignment, and average blank/silence posterior output $\mathbb{E}p(\epsilon \mid x)$. $N_{\text{words}} \in \{1, 2, 3\}$ and fixed $N_{\text{rep}} = 2$, $r_{\text{sil}} = 0.3$. The reference alignment has 21% silence. TSE is in number of frames. All detailed definitions are in Appendix A.4.2. An extended version of this table is Appendix Table 8.

| Posterior Scale $\alpha$ | Prior Scale $\beta$ | Transition Scale $\gamma$ | LER [%] $\mu$ | $\sigma$ | Fw. CE $\mu$ | $\sigma$ | $\mathbb{E}p(\epsilon \mid x)$ [%] $\mu$ | $\sigma$ | TSE $\mu$ | $\sigma$ |
|---|---|---|---|---|---|---|---|---|---|---|
| 0.5 | 0.2 | 0.0 | 5.9 | 4.7 | 0.51 | 0.06 | 23 | 4 | 0.3 | 0.1 |
| 0.5 | 0.3 | 0.2 | 4.7 | 4.4 | 0.50 | 0.10 | 18 | 4 | 0.3 | 0.1 |
| 0.5 | 0.0 | 0.5 | 5.0 | 5.0 | 0.45 | 0.06 | 20 | 3 | 0.2 | 0.1 |
| 0.5 | 0.0 | 0.0 | 6.7 | 6.5 | 0.60 | 0.13 | 30 | 3 | 0.5 | 0.1 |
| 1.0 | 0.0 | 0.0 | 63.7 | 14.3 | 3.57 | 0.63 | 11 | 20 | 1.3 | 0.3 |

Table 2: Comparing **phoneme-based HMM/CTC** on Switchboard 300h. Overview of time stamp error (TSE) on word boundaries of the alignments with respect to a GMM alignment, the percentage of silence (Si) in HMM and blank (B) in CTC, as well as the average phoneme duration (Phon). We show different modeling approach variants for Switchboard 300h using label posterior, prior, and transition scales, $\alpha$, $\beta$, and $\gamma$ respectively. All decoding experiments use a 4gram LM.

| Model | Posterior Scale $\alpha$ | Prior Scale $\beta$ | Transition Scale $\gamma$ | Align model on SWB 300h TSE [ms] | Si/B [%] | Phon.[ms] | WER [%] HUB5'00 | HUB'01 |
|---|---|---|---|---|---|---|---|---|
| GMM | 1.0 | 0.0 | 1.0 | 0.0 | 25.1 | 86.5 | 18.9 | - |
| CTC | | | 0.0 | 89.5 | 65.6 | 40.0 | 12.8 | 11.8 |
| HMM | 0.7 | 0.0 | 0.3 | 73.0 | 38.0 | 71.2 | 12.4 | 11.6 |
| | | | 0.0 | 107.6 | 22.5 | 89.3 | 12.3 | 11.5 |
| | 0.5 | 0.3 | | 350.0 | 2.6 | 112.1 | 12.8 | 11.9 |
| | 0.7 | 0.1 | 0.1 | 139.1 | 10.0 | 103.5 | 12.2 | 11.5 |

We use fixed normalized transition probabilities with four values for speech and non-speech forward/loop. We make use of the knowledge of 80ms average duration for phonemes based on our best GMM alignments and therefore choose a loop/forward probability of $0.5$ when using 40ms downsampling. The silence transition values are estimated based on the sentence begin/end silence frames averaged on all utterances, for roughly $0.04$ forward probability. We considered three different prior estimation method: (1) fixed and estimated based on transcriptions (Raissi et al., 2022), (2) averaged over time frames of the current sequence (3) or similarly averaged over the whole batch. A comparison between the different models is shown in Appendix Table 12. We use the sequence-based estimation for our experiments.

**Switchboard (Godfrey et al., 1992)** The comparison of different modeling approaches for phoneme-based HMM is shown in Table 2. We see that the use of prior leads to higher TSE, especially when no transition model is used. The approach with lowest TSE and WER avoids the use of prior during training but makes use of the transition model. Regarding the duration model, none of the approaches could match the GMM statistics in terms of silence percentage and average phoneme duration. The model using both transition model and prior obtains the best WER, however due to the silence prior correction the alignment suffers the lack of silence frame and therefore has higher TSE. Similar observations have been done in prior work (Raissi et al., 2022).

**LibriSpeech (Panayotov et al., 2015)** As shown in Appendix Table 13, for this task we observe similar results for the use of prior in terms of high TSEs. The best WER and TSE combination for HMM also in this case uses only the transition model. This result is consistent not only with

Table 3: Results using **normalized gradient** for the CTC model using SPM10k vocab on LibriSpeech. TSE is w.r.t. the same GMM alignment as in Table 13. We penalize the blank probability and divide by prior for obtaining the alignments (for TSE / sil. ratio).

| $\overline{v}_{\min}$ | $\overline{v}_{\max}$ | $\mathbb{E}_t v_t$ **Est.** | **TSE** [ms] | | **Sil. ratio** [%] | **WER** [%] | |
|---|---|---|---|---|---|---|---|
| | | | LR | Center | | dev-other | test-other |
| Reference GMM alignment | | | 0 | 0 | 18.0 | - | - |
| 1.0 | 1.0 | - | 68.2 | 52.0 | 21.8 | 5.77 | 6.03 |
| 0.5 | 1.1 | Batch | 78.9 | 66.7 | 22.7 | 5.71 | 5.87 |
| 0.1 | 1.1 | Batch | 154.7 | 151.9 | 22.3 | 6.21 | 6.55 |
| 0.5 | 1.1 | Seq. | 70.7 | 54.6 | 23.7 | 5.83 | 5.91 |

Table 4: Comparison of **blank separation** and **normalized gradient** ($\overline{v}_{\min} = 0.5$, $\overline{v}_{\max} = 1.1$) on CTC models with varying vocabularies. TSE is w.r.t. the same GMM alignment as in Tables 2 and 3. The reference GMM alignment has 18.0% silence ratio. We penalize the blank probability and divide by prior for obtaining the alignments (for TSE / sil. ratio).

| **Vocab.** | **Method** | **TSE** [ms] | | **Sil. ratio** [%] | **WER** [%] | |
|---|---|---|---|---|---|---|
| | | LR | Center | | dev-other | test-other |
| SPM 512 | - | 58.2 | 47.8 | 13.5 | 5.97 | 6.21 |
| | Blank sep | 58.7 | 50.5 | 16.8 | 6.02 | 6.04 |
| SPM 10k | - | 68.2 | 52.0 | 21.8 | 5.77 | 6.03 |
| | Blank sep | 84.4 | 75.4 | 26.6 | 5.73 | 6.02 |
| | Normed grad, seq. | 70.7 | 54.6 | 23.7 | 5.83 | 5.91 |
| | Normed grad, batch | 78.9 | 66.7 | 22.7 | 5.71 | 5.87 |
| | Normed grad, batch + blank sep | 72.9 | 58.8 | 28.8 | 5.73 | 6.08 |
| BPE 10k | - | 66.2 | 56.3 | 22.5 | 6.18 | 6.35 |
| | Blank sep | 72.5 | 65.3 | 26.7 | 5.98 | 6.13 |

Switchboard experiments. This model has also the best WER. However, the best TSE in this set of experiments is obtained by the CTC model. We also observed a slight difference in use of the label and transition scales for LibriSpeech task.

## 7.2 NORMALIZED GRADIENT

Results with our CTC model on LibriSpeech with the normalized gradient training criterion are in Table 3. While we can get some small improvement over the baseline in terms of WER, we also see that it is sensitive to the clamping values $\overline{v}_{\min}$ and $\overline{v}_{\max}$. There is only a small difference between batch-based or sequence-based estimation of $\overline{v} = \mathbb{E}_t v_t$, maybe batch-based being slightly better. Unexpectedly, there does not seem to be any improvement in terms of alignment quality (TSE). Also, in terms of convergence rate, there was no difference (see Appendix Figure 2).

## 7.3 BLANK SEPARATION IN CTC

We tested different blank penalties and prior scales to obtain the alignments (see Appendix A.7, Appendix Table 14). Here we present the best variant using blank penalty $-10$ and with prior scale 1. Results when separating the blank symbol in comparison to the baseline and also to normalized gradients can be seen in Table 4. Unexpectedly, there does not seem to be any improvement in terms of alignment quality (TSE) and the baseline has the best TSE. The improvement in terms of WER is small, but there seem to be a consistent improvement in most cases. Note, in terms of convergence rate, there was no difference here (see Appendix Figure 2).

## 7.4 ALIGNMENTS VIA GRADIENTS

We collect our CTC gradient-based alignment results in Table 5. GradScoreExt performs a bit better than GradScore for SPM/BPE 10k, but slightly worse for SPM 512. Compared to the CTC forced alignment quality (Table 4), we see a small improvement in TSE in most cases except of

Table 5: TSE for CTC gradient-based alignments. These are the same models as in Table 4. We use $p = 0.1$ for the grad norm. We use prior but no blank penalty for all but the blank-sep. models.

| Vocab. | Method | Align Variant | TSE [ms] | | Sil. ratio [%] |
|---|---|---|---|---|---|
| | | | LR | Center | |
| SPM 512 | - | GradScore | 76.6 | 60.2 | 17.4 |
| | | GradScoreExt | 77.6 | 61.0 | 14.0 |
| | Blank sep | GradScore | 73.9 | 58.8 | 18.5 |
| | | GradScoreExt | 75.9 | 60.5 | 13.6 |
| SPM 10k | - | GradScore | 69.9 | 51.1 | 21.1 |
| | | GradScoreExt | 67.9 | 50.2 | 15.9 |
| | Blank sep | GradScore | 77.1 | 58.4 | 24.1 |
| | | GradScoreExt | 72.5 | 55.7 | 15.3 |
| | Normed grad | GradScore | 70.7 | 53.5 | 20.4 |
| | | GradScoreExt | 69.2 | 53.0 | 15.5 |
| BPE 10k | - | GradScore | 72.9 | 55.3 | 21.2 |
| | | GradScoreExt | 71.3 | 54.7 | 16.2 |
| | Blank sep | GradScore | 72.7 | 55.3 | 23.8 |
| | | GradScoreExt | 67.3 | 51.1 | 15.2 |

Table 6: TSE for AED gradient-based alignments. SPM10k vocab. The AED model has 4.98% and 5.49% WER on dev-other/test-other respectively. The ref. GMM alignment has 18.0% sil. ratio.

| Align Variant | TSE [ms] | | Sil. ratio [%] |
|---|---|---|---|
| | LR | Center | |
| GradScore | 66.3 | 50.5 | 23.7 |
| GradScoreExt | 64.7 | 50.3 | 14.9 |

SPM 512. The alignment quality seems to be more robust in comparison to CTC forced alignments, where it can vary widely depending on the conditions like vocabulary size and scales.

We also apply the method on an AED model, and show the AED gradient-based alignment results in Table 6. The AED model is expectedly a bit better than the CTC model (5.5% on test-other vs. 6.0% on test-other). In comparison to CTC, the alignment quality seems to be better here in terms of TSE. Again GradScoreExt performs a bit better than GradScore. We also test a hybrid AED/CTC model in Appendix A.8.

## 8  CONCLUSIONS

The use of synthetic data provides us with a very useful tool. We find that the framerate and amount of noise play a crucial role on the training dynamics. The prior is most important for higher framerates and not needed for lower framerates. In the noisy synthetic case, the combination of posterior, prior and transition model works best. For real data, use of prior results in alignment quality degradation and use of transition model together with posterior is sufficient.

The separation of the blank symbol and the normalized gradient do not improve the alignment quality (TSE) but they slightly improve the WER. The blank separation allows for faster greedy decoding and faster framewise training. Our novel gradient-based method to find an alignment improves the alignment quality in case of larger vocabularies. The alignment quality also seems to be more robust in comparison to CTC forced alignments.

Forced alignments using smaller vocabularies and also phoneme-based models, specifically small models, or even GMMs, still provides the best alignments though, but with potentially much worse WER. With the gradient-based alignment method, the alignment quality interestingly seems to correlate much better with WER.

## 9    REPRODUCIBILITY STATEMENT

All the code for all experiments, including the whole setup pipeline with dataset preparation, training and recognition will be published.

We further list all relevant details about our setup, including software and hardware, in Section 6 and Appendix A.4.

The used hardware and software should be easily available to everyone.

Thus, it should not be any problem to reproduce our results, within the limit of randomness in the used training algorithms, and small differences when using different hardware or different software versions.

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

# A  APPENDIX

## CONTENTS

## A.1   RELATED WORK

The work by Zeyer et al. (2017; 2021); Raissi et al. (2022); Chen et al. (2023); Huang et al. (2024); Raissi et al. (2024) is very related in that they also investigate the alignment behavior of CTC or HMM, and improve by using a prior in training. The way how the prior is estimated differs: E.g. Huang et al. (2024) reestimates the prior every epoch. Initially it is uniform, and then the model softmax output average. Zeyer et al. (2017) estimates the prior by a running exponential average of the model softmax output. Raissi et al. (2022) estimates the prior from the transcriptions and keeps it fixed. In this work here, we mostly use a prior estimated based on the current sequence, but we compare several variants (see Tables 7 and 12).

The work by Zeyer et al. (2021); Raissi et al. (2022); Zhao & Bell (2023); Raissi et al. (2024); Zhao & Bell (2024) studies the influence of label topology, e.g. CTC with the special blank, or the standard HMM, or other variations. Here we also compare different variants, specifically standard HMM and CTC, but our synthetic framework also allows to study any other variant. Zhao & Bell (2023) notes that the frame rate also determines what label topology is optimal. We also find that the frame rate is very important on what training criterion works best, e.g. with prior or without.

The work by Huang et al. (2024); Rousso et al. (2024) compares the alignment quality of different model types.

Deep learning algorithms can fare poorly when the training dataset suffers from heavy class-imbalance. The blank or silence label in speech recognition is very imbalanced compared to the other labels. There is a lot of related work on how the problems and potential solutions when there is class-imbalance (Johnson & Khoshgoftaar, 2019; Chen et al., 2024). Both the normalized gradient method and the blank separation modeling are closely related to other variants of class-balancing the loss such as (Lin et al., 2018; Cao et al., 2019).

The work by Schmitt et al. (2024) introduces the same method to extract alignments from the gradients w.r.t. the inputs. However, this was done only for AED models. Here it is extended for CTC models and further improved.

## A.2   DERIVATION OF GRADIENTS FOR SEPARATED BLANK IN CTC

Recall the definition of separated blank in the CTC model:

$$\log p'_{\mathcal{Y}}(y \mid h_t) = \begin{cases} \log \sigma(z_\epsilon), & y = \epsilon \\ \log \sigma(-z_\epsilon) + \log \operatorname{softmax}_{\mathcal{A}}(z_{\mathcal{A}})_y, & y \in \mathcal{A} \end{cases} \tag{28}$$

For $\log \sigma$, we get the gradient:

$$\log \sigma(x) = -\log(1 + \exp(-x)) \tag{29}$$

$$\nabla_x \log \sigma(x) = -\frac{1}{1 + \exp(-x)} \cdot \exp(-x) \cdot (-1) \tag{30}$$

$$= \frac{1}{1 + \exp(x)} \tag{31}$$

$$= \sigma(-x) \tag{32}$$

$$= 1 - \sigma(x) \tag{33}$$

$$\nabla_x \log \sigma(-x) = -\frac{1}{1 + \exp(-x)} \cdot \exp(-x) \tag{34}$$

$$= -\frac{1}{1 + \exp(-x)} \tag{35}$$

$$= -\sigma(x) \tag{36}$$

For the softmax, we get the gradient:

$$\nabla_{z_j} \log \text{softmax}(z)_i = \delta_{i=j} - \text{softmax}(z)_j \tag{37}$$

with the Kronecker delta $\delta$.

Putting it together:

$$\nabla_{z_j} \log p'_{\mathcal{Y}}(y{=}i \mid h_t) = \begin{cases} 1 - \sigma(z_\epsilon), & j = \epsilon, i = \epsilon, \\ -\sigma(z_\epsilon), & j = \epsilon, i \neq \epsilon, \\ 0, & j \neq \epsilon, i = \epsilon, \\ \delta_{i=j} - \text{softmax}_{\mathcal{A}}(z_{\mathcal{A}})_j, & j \neq \epsilon, i \neq \epsilon \end{cases} \tag{38}$$

I.e., for some given soft alignment / target probability distribution $\upsilon \in \mathbb{R}^{\mathcal{Y}}$, we get:

$$\nabla z_j \sum_i \upsilon_i \log p'_{\mathcal{Y}}(y{=}i \mid h_t) = \begin{cases} (1 - \sigma(z_\epsilon)) \cdot \upsilon_\epsilon - \sigma(z_\epsilon) \cdot (\sum_{i \neq \epsilon} \upsilon_i), & j = \epsilon, \\ \upsilon_j - \text{softmax}_{\mathcal{A}}(z_{\mathcal{A}})_j \cdot (1 - \upsilon_\epsilon), & j \neq \epsilon \end{cases} \tag{39}$$

$$= \begin{cases} \upsilon_\epsilon - \sigma(z_\epsilon), & j = \epsilon, \\ (\frac{\upsilon_j}{1 - \upsilon_\epsilon} - \text{softmax}_{\mathcal{A}}(z_{\mathcal{A}})_j) \cdot (1 - \upsilon_\epsilon), & j \neq \epsilon \end{cases} \tag{40}$$

In comparison, for the full softmax (not separated blank), we get:

$$\nabla z_j \sum_i \text{stopgrad}(\upsilon_i) \cdot \log p_{\mathcal{Y}}(y{=}i \mid h_t) = \upsilon_j - \text{softmax}_{\mathcal{Y}}(z)_j \tag{41}$$

## A.3 TRAINING SCORES

We plot the CTC training scores in Figure 2 on Librispeech with SPM10k vocab. There don't seem to be any difference.

## A.4 EXPERIMENTAL SETUP DETAILS

### A.4.1 CORPORA

**Switchboard** We use the 300h Switchboard-1 Release 2 (LDC97S62) (Godfrey et al., 1992). We evaluate our models on Switchboard and CallHome subsets of Hub5'00 (LDC2002S09), the three subsets of Hub5'01 (LDC2002S13).

**LibriSpeech** For a larger set of experiments we considered the 960h LibriSpeech (Panayotov et al., 2015), with evaluations on dev-other and test-other.

### A.4.2 METRIC DEFINITIONS

**Label-error-rate (LER) / Word-error-rate (WER)** The label-error-rate (LER) and word-error-rate (WER), also called edit distance or Levenshtein distance, is given by

$$\text{LER} = \frac{N_{\text{sub}} + N_{\text{ins}} + N_{\text{del}}}{N_{\text{ref labels}}},$$

where $N_{\text{sub}}, N_{\text{ins}}, N_{\text{del}}$ refer to number of substitutions, insertions and deletions, and represent the minimum amount of edits needed to perform to transform the recognized label sequence into the reference label sequence. For speech recognition, the WER (calculated on word-level) is one of the most important metrics. It does not measure the alignment quality in any way though, and a model can have a good WER but bad alignment quality (CTC models often have this), or a model can have good alignment quality but bad WER (e.g. a GMM).

**Time-stamp-error (TSE)** The TSE is the sum of distances between start and end frames for each word w.r.t. some reference alignment (always from a GMM here) divided by number of words times 2. This is calculated over some set of sequences.

$$\text{TSE} = \frac{\sum_w |t_{w,\text{start,ref}} - t_{w,\text{start,model}}| + |t_{w,\text{end,ref}} - t_{w,\text{end,model}}|}{2 \cdot N_{\text{words}}}$$

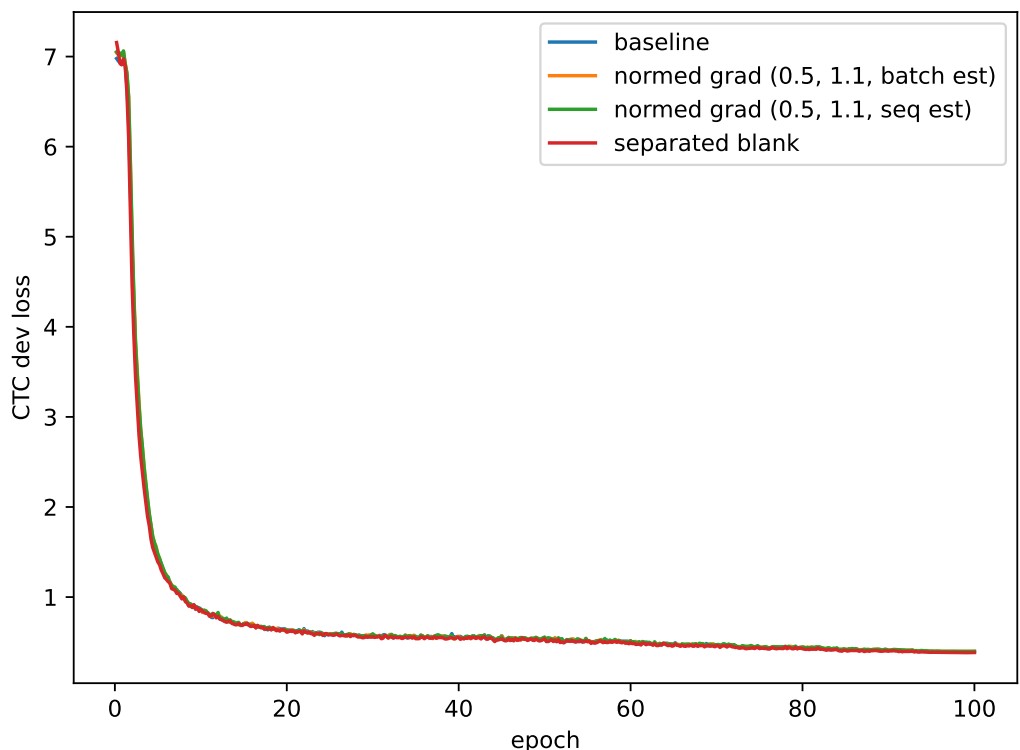

Figure 2: CTC training scores on Librispeech dev. Using SPM10k vocab.

For the synthetic experiments, $t$ is in terms of frame index, while for all the real data experiments, we use the real time (seconds or milliseconds).

The TSE can be calculated also when the model operates on BPE and the reference alignment is on phonemes, as long as both allow to determine the word boundaries.

**Framewise cross entropy (Fw. CE)** Given a reference alignment, i.e. for some label sequence $a_1^S$, the alignment label sequence $y_1^T$, the framewise cross entropy (fw. CE) for CTC models or HMMs is defined as

$$L_{\text{CE}} = -\sum_{t=1}^{T} \log p(y_t \mid h_t).$$

For hybrid NN-HMM, it is common to also use this criterion in training, based on a given external alignment (which often comes from a GMM). But even when this criterion is not used for training, it provides a measure on how close the model is to this alignment.

Thus it's another alternative to TSE (when the vocab matches, i.e. the given alignment can be evaluated directly like that; when the alignment is on phonemes, and the model operates on BPE, this does not work).

**Average blank/silence posterior output** $\mathbb{E}p(\epsilon \mid x)$ The average blank/silence posterior output is given by

$$\mathbb{E}p(\epsilon \mid x) = \frac{1}{T} \sum_t p(y_t{=}\epsilon \mid x).$$

This is calculated over a set of sequences. This probability number indicates how much the model prefers silence or blank. A realistic amount of silence is 20%. If we get a much larger number for the average amount of blank (e.g. 80%), it means we have peaky behavior.

### A.4.3 SOFTWARE

We use PyTorch 2.1.0 Paszke et al. (2019) for the experiments on synthetic data and for the subword-based models, and TensorFlow 2.3[10] Abadi et al. (2015) for the phoneme-based models. We use Slurm Yoo et al. (2003) for the cluster job queueing.

### A.4.4 HARDWARE

We use two types of GPUs: Nvidia 1080 or 2080. The phoneme-based models were trained using a single GPU, and the subword-based models are always trained with 4 GPUs distributed training.

The experiments on synthetic data are mostly executed on a single Apple M1 Pro CPU.

### A.4.5 SYNTHETIC FRAMEWORK

We define a set of words and their mapping to a sequence of labels in $\mathcal{A}$. In most of the experiments, we use the artificial words "helo", "world", "howe", "are", "you", and map each word to their characters, so we end up with $|\mathcal{A}| = 10$ possible labels. The alignment labels augment those by blank or silence: $\mathcal{Y} = \mathcal{A} \cup \{\epsilon\}$. To sample a reference alignment: First sample the number of words $N_{\text{words}}$ from a uniform distribution $[N_{\text{words}_{\min}}, N_{\text{words}_{\max}}] \subset \mathbb{N}$. Then map the word to the sequence of labels in $\mathcal{A}$. Then, for each label, sample its number of repetitions from a uniform distribution $[N_{\text{rep}_{\min}}, N_{\text{rep}_{\max}}] \subset \mathbb{N}$. Next, sample the silence factor $r_{\text{sil}}$ from a uniform distribution $[R_{\text{sil}_{\min}}, R_{\text{sil}_{\max}}] \subset \mathbb{R}$. Given the length of the sequence counting the repetitions as $T_{\mathcal{A}}$, this factor tells how much silence to add as $T_\epsilon = r_{\text{sil}} \cdot T_{\mathcal{A}}$. The silence frames $\epsilon$ are added uniformly before/after words, so at $N_{\text{words}} + 1$ possible positions. This procedure will construct a ground truth alignment label sequence $y_1^T \in \mathcal{Y}^T$ with $T = T_{\mathcal{A}} + T_\epsilon$.

We use the input feature dimension $D = \mathcal{Y}$. The input features $x_1^T \in \mathbb{R}^{T \times D}$ are simply constructed from the alignment label sequence by the one-hot encoding[11] with optional noise $\xi_i \in \mathcal{N}(0, 1)$ and noise scale $\sigma_\xi$:

$$x_{t,i} = (1 - \sigma_\xi) \cdot \delta_{i=y_t} + \sigma_\xi \cdot \xi_i. \tag{42}$$

From this construction, there is a trivial optimal mapping (with $\sigma_\xi = 0$) from the input features to the target probability distribution $p(y \mid x)$:

$$p_{\text{opt}}(y{=}i \mid x_t) = x_{t,i} \tag{43}$$

With no noise, this is a global optimum to most of the training criteria variations (especially for the standard CTC), and this model will also provide a perfect alignment.

For the trained model, we use either the HMM label topology, where $\epsilon$ is only allowed before/after words, which also matches how we construct the ground truth alignment, or we use the CTC label topology, where $\epsilon$ is allowed anywhere.

### A.4.6 PHONEME-BASED MODELS

The phoneme-based experiments for HMM and CTC are carried out on Switchboard and LibriSpeech. The speech signal is extracted using a 25ms window with a 10ms shift, yielding Gammatone filterbank features with dimensions of 40 (Schlüter et al., 2007). All Conformer models use a downsampling of factor 4. SpecAugment is applied across all models (Park et al., 2019). All encoder architectures consist of a 12-layer Conformer encoder with 75 million parameters (Gulati et al., 2020). All models are trained for 50 epochs on Switchboard and 25 epochs on LibriSpeech. We use one cycle learning rate schedule (OCLR) up to peak LR of 6e-4 over 90% of the training epochs, followed by a linear decrease to 1e-6 (Smith & Nicholay, 2019; Zhou et al., 2022). An Adam optimizer Nesterov momentum, together with optimizer epsilon of 1e-8 are used (Kingma & Ba, 2015; Dozat, 2016).

---

[10]Yes, this is old...

[11]This is similar to the work in Zeyer et al. (2021).

### A.4.7  SUBWORD-BASED MODELS

We use byte-pair-encoding (BPE) (Sennrich et al., 2016) or sentence-piece models (SPM) with unigram LM (Kudo, 2018) as subword units. Our CTC model uses a Conformer encoder (Gulati et al., 2020).

We use Adam (Kingma & Ba, 2015) with decoupled weight decay (AdamW) (Loshchilov & Hutter, 2019). In multi-GPU training, we average the parameters every 100 steps. We use one cycle learning rate schedule (OCLR) up to peak LR of 1e-4 over 90% of the training epochs, followed by a linear decrease to 1e-6 (Smith & Nicholay, 2019).

We use SpecAugment (Park et al., 2019), speed perturbation, dropout (Hinton et al., 2012), we sample different subword segmentations, and we use an auxiliary AED loss (but without using more data) (Hentschel et al., 2024).

### A.5  EXPERIMENTS ON SYNTHETIC DATA

**Effect of prior for simple FFNN, high framerate**  See Table 7.

Table 7: Comparing the effect of the prior and different posterior/prior scales $\alpha/\beta$, on synthetic data without noise, using a simple FFNN, HMM label topology. No transition model here. The experiment is repeated over 10 random seeds to measure the mean $\mu$ and standard deviation $\sigma$. We use the fixed $N_{\text{words}} = 1$, $N_{\text{rep}} = 10$ and $r_{\text{sil}} = 1.0$, thus the reference alignment has always 50% silence. We provide the label-error-rate (LER) as an indicator of the performance of the model. We calculate the framewise CE w.r.t. the reference alignment, and the average blank/silence posterior output $\mathbb{E}p(\epsilon \mid x)$. TSE is in number of frames.

| Prior | | | Posterior | LER [%] | | Fw. CE | | $\mathbb{E}p(\epsilon \mid x)$ [%] | | TSE | |
|---|---|---|---|---|---|---|---|---|---|---|---|
| Type | Stop Grad | $\beta$ | $\alpha$ | $\mu$ | $\sigma$ | $\mu$ | $\sigma$ | $\mu$ | $\sigma$ | $\mu$ | $\sigma$ |
| Posterior avg. | Yes | 0.5 | 0.5 | 0.0 | 0.0 | 0.00 | 0.00 | 50 | 0 | 0.0 | 0.0 |
| | | 1.0 | 1.0 | 0.5 | 1.1 | 0.05 | 0.10 | 50 | 0 | 0.1 | 0.2 |
| | No | 0.5 | 0.5 | 33.8 | 26.0 | 1.79 | 1.94 | 45 | 15 | 0.4 | 1.2 |
| Static | - | 0.5 | 0.5 | 108.2 | 10.1 | 11.39 | 0.74 | 0 | 0 | 19.5 | 0.0 |
| | | 0.1 | 0.5 | 8.7 | 20.8 | 0.39 | 0.86 | 52 | 3 | 0.6 | 1.2 |
| | | 0.03 | 0.1 | 11.8 | 11.0 | 0.28 | 0.11 | 50 | 8 | 0.9 | 1.4 |
| - | - | 0 | 0.5 | 36.9 | 25.3 | 2.19 | 1.78 | 64 | 13 | 5.7 | 5.1 |
| | | | 1.0 | 65.9 | 32.0 | 5.28 | 4.26 | 68 | 27 | 11.2 | 5.5 |

**Comparing posterior/prior/transition scales in the presence of noise**  See Table 8.

**Comparing HMM vs. CTC label topology**  How does HMM and CTC label topology compare in training? Note, the difference between HMM and CTC is just where you allow $\epsilon$ (which is called "blank" for CTC and is treated as silence for HMM). See Figures 3 and 4 for the finite state automata (FSA) of HMM and CTC label topology for some example sequence. How does this difference influence the training? Here we focus on standard case of the CTC training criterion with posterior scale $\alpha = 1$ and no prior and no transition model (prior and transition scale $\beta = \gamma = 0$). We compare several cases in Table 9.

We note that the results here depend a lot on the specific settings. Specifically, the model size, the dataset distribution (amount of noise, amount of frames per label, etc.), the amount of training, the training hyper parameters all influence the results. Already the training dynamics will vary a lot, and thus also the alignment behavior (very peaky or not). Depending on this, it's not always the case that the CTC label topology is better for these scales ($\alpha = 1, \beta = \gamma = 0$). There is more ongoing work on getting a more complete picture of all these aspects, but this is going beyond the current presented work here.

**Effect of the synthetic dataset distribution**  See Table 10.

Table 8: Extended version of Table 1. Comparing the posterior/prior/transition scales with HMM label topology in the presence of noise ($\sigma_\xi = 0.5$) with a 2-layer BLSTM posterior model with 20 dimensions in each direction, and higher batch size 100. We use the posterior average as prior with stop gradient. The experiment is repeated over 10 random seeds to measure the mean $\mu$ and standard deviation $\sigma$. We provide the label-error-rate (LER) as an indicator of the performance of the model. We calculate the framewise CE w.r.t. the reference alignment, and the average blank/silence posterior output $\mathbb{E}p(\epsilon \mid x)$. We use $N_{\text{words}} \in \{1, 2, 3\}$ and fixed $N_{\text{rep}} = 2$, $r_{\text{sil}} = 0.3$. The reference alignment has 21% silence. TSE is in number of frames.

| Posterior Scale $\alpha$ | Prior Scale $\beta$ | Transition Scale $\gamma$ | LER [%] $\mu$ | $\sigma$ | Fw. CE $\mu$ | $\sigma$ | $\mathbb{E}p(\epsilon \mid x)$ [%] $\mu$ | $\sigma$ | TSE $\mu$ | $\sigma$ |
|---|---|---|---|---|---|---|---|---|---|---|
| 0.5 | 0.5 | 0.0 | 10.5 | 5.2 | 0.54 | 0.09 | 15 | 2 | 0.3 | 0.1 |
| 0.5 | 0.2 | 0.0 | 5.9 | 4.7 | 0.51 | 0.06 | 23 | 4 | 0.3 | 0.1 |
| 0.5 | 0.5 | 0.1 | 9.4 | 5.8 | 0.55 | 0.05 | 15 | 3 | 0.3 | 0.1 |
| 0.5 | 0.5 | 0.2 | 7.7 | 5.4 | 0.52 | 0.05 | 16 | 2 | 0.3 | 0.0 |
| 0.5 | 0.5 | 0.3 | 7.8 | 5.7 | 0.53 | 0.08 | 15 | 3 | 0.3 | 0.1 |
| 0.5 | 0.3 | 0.2 | 4.7 | 4.4 | 0.50 | 0.10 | 18 | 4 | 0.3 | 0.1 |
| 0.5 | 0.0 | 0.5 | 5.0 | 5.0 | 0.45 | 0.06 | 20 | 3 | 0.2 | 0.1 |
| 0.5 | 0.0 | 0.2 | 7.8 | 9.7 | 0.52 | 0.15 | 24 | 2 | 0.3 | 0.2 |
| 0.5 | 0.0 | 0.1 | 7.2 | 5.0 | 0.57 | 0.08 | 26 | 4 | 0.4 | 0.1 |
| 0.5 | 0.0 | 0.0 | 6.7 | 6.5 | 0.60 | 0.13 | 30 | 3 | 0.5 | 0.1 |
| 1.0 | 0.0 | 0.0 | 63.7 | 14.3 | 3.57 | 0.63 | 11 | 20 | 1.3 | 0.3 |

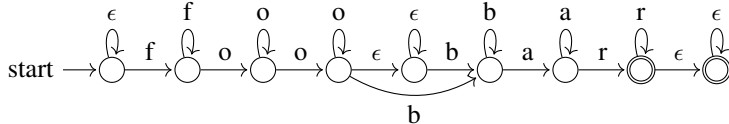

Figure 3: FSA for HMM label topology for characters in "foo bar"

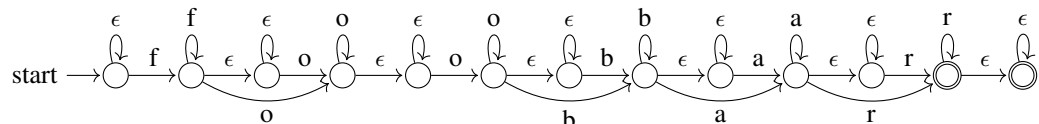

Figure 4: FSA for CTC label topology for characters in "foo bar"

Table 9: Comparing HMM vs. CTC label topology in the presence of noise ($\sigma_\xi = 0.5$) with a 2-layer BLSTM posterior model with varying number of dimensions, and higher batch size 100, with posterior scale $\alpha = 1$ and no prior and no transition model (prior and transition scale $\beta = \gamma = 0$). The experiment is repeated over 10 random seeds to measure the mean $\mu$ and standard deviation $\sigma$. We provide the label-error-rate (LER) as an indicator of the performance of the model. We calculate the framewise CE w.r.t. the reference alignment, and the average blank/silence posterior output $\mathbb{E}p(\epsilon \mid x)$. We use $N_{\text{words}} \in \{1, 2, 3\}$ and fixed $N_{\text{rep}} = 2$, $r_{\text{sil}} = 0.3$. The reference alignment has 21% silence. TSE is in number of frames.

| LSTM # Dim. | Label Topology | LER [%] $\mu$ | $\sigma$ | Fw. CE $\mu$ | $\sigma$ | $\mathbb{E}p(\epsilon \mid x)$ [%] $\mu$ | $\sigma$ | TSE $\mu$ | $\sigma$ |
|---|---|---|---|---|---|---|---|---|---|
| 20 | HMM | 63.7 | 14.3 | 3.57 | 0.63 | 11 | 20 | 1.3 | 0.3 |
|  | CTC | 14.2 | 8.2 | 0.41 | 0.07 | 39 | 4 | 1.3 | 0.5 |
| 100 | HMM | 14.8 | 9.7 | 2.04 | 0.56 | 41 | 7 | 1.1 | 0.1 |
|  | CTC | 5.5 | 4.7 | 0.21 | 0.05 | 29 | 3 | 0.5 | 0.6 |

Table 10: Comparing the effect of the synthetic dataset distribution without noise, using a simple FFNN, HMM label topology. No transition model here. We use the posterior average as prior with stop gradient. The experiment is repeated over 10 random seeds to measure the mean $\mu$ and standard deviation $\sigma$. We provide the label-error-rate (LER) as an indicator of the performance of the model. We calculate the framewise CE w.r.t. the reference alignment, and the average blank/silence posterior output $\mathbb{E}p(\epsilon \mid x)$. TSE is in number of frames.

| Dataset | | | Prior Scale | Posterior Scale | LER [%] | | Fw. CE | | $\mathbb{E}p(\epsilon \mid x)$ [%] | | TSE | |
|---|---|---|---|---|---|---|---|---|---|---|---|---|
| Num frames / label | Silence [%] factor | ratio | $\beta$ | $\alpha$ | $\mu$ | $\sigma$ | $\mu$ | $\sigma$ | $\mu$ | $\sigma$ | $\mu$ | $\sigma$ |
| 10 | 100 | 50 | 0.5 | 0.5 | 0.0 | 0.0 | 0.00 | 0.00 | 50 | 0 | 0.0 | 0.0 |
| | 20 | 17 | 0.5 | 0.5 | 0.0 | 0.0 | 0.00 | 0.00 | 17 | 0 | 0.0 | 0.0 |
| | | | 0.0 | 1.0 | 45.6 | 29.2 | 4.90 | 4.01 | 51 | 26 | 7.9 | 6.0 |
| 2 | | 14 | 0.5 | 0.5 | 1.5 | 3.2 | 0.09 | 0.05 | 15 | 3 | 0.0 | 0.0 |
| | | | 0.0 | 1.0 | 0.0 | 0.0 | 0.00 | 0.00 | 14 | 0 | 0.0 | 0.0 |
| | 50 | 19 | 0.0 | 1.0 | 0.0 | 0.0 | 0.00 | 0.00 | 19 | 0 | 0.0 | 0.0 |

**Perfect initialization**    Note that in all cases, we can initialize the model parameters in a way that we get as close as we want to perfect alignment behavior. The model needs to be initialized in such a way that it performs a scaled identity function. This is possible with all the studied models.

A.6   PHONEME-BASED MODELS

See Table 11 for the effect of the number of epochs.

Table 11: Evaluation for Conformer based HMM with only transition model and no prior trained from scratch for 25 and 100 epochs on LibriSpeech 960h and decoded using 4gram LM.

| Model | Epochs | WER [%] | |
|---|---|---|---|
| | | dev-other | test-other |
| HMM | 25 | 6.6 | 7.1 |
| | 100 | 5.9 | 5.8 |

See Table 12 for the effect of the type of prior.

Table 12: Effect of use of different prior for from-scratch trained Conformer based HMM with only prior and with no transition model. The model is trained from scratch Switchboard 300h for 50 epochs and evaluated on Hub5'00 using 4-gram LM. The fixed prior is estimated on the transcriptions.

| Model | Prior | dev-other [%] |
|---|---|---|
| | Fixed | 13.3 |
| HMM | Batch | 13.3 |
| | Seq. | 12.8 |

See Table 13 for a comparison of different models and label topologies (GMM, CTC, HMM) and different posterior, prior and transition scales.

A.7   BLANK SEPARATION IN CTC

This is an extension to Section 7.3.

**Multiple variants to obtain an alignment**    We tested different blank penalties (just adding a constant bias to the logits of blank) and prior scales on the influence of the alignment quality when doing forced alignment with the CTC model. The results are in Table 14. As expected, without prior, without blank penalty, the CTC model is very peaky. The best TSE is obtained with a combination of both: Blank penalty shift -10 and prior scale 1.

Table 13: Comparing **phoneme-based HMM/CTC** on LibriSpeech 960h. Overview of time stamp error (TSE) on word boundaries of the alignments with respect to a GMM alignment, the percentage of silence (Si) in HMM and blank (B) in CTC, as well as the average phoneme duration (Phon). We show different modeling approach variants for LibriSpeech 960h using label posterior, prior, and transition scales, $\alpha$, $\beta$, and $\gamma$ respectively. All decoding experiments use a 4gram LM. Similar experiments as presented in Table 2 for Switchboard.

| Model | Posterior Scale $\alpha$ | Prior Scale $\beta$ | Transition Scale $\gamma$ | Align model on train 960h | | | WER [%] | |
|---|---|---|---|---|---|---|---|---|
| | | | | TSE [ms] | Si/B [%] | Phon.[ms] | HUB5'00 | HUB'01 |
| GMM | 1.0 | 0.0 | 1.0 | 0.0 | 17.5 | 85.0 | 19.8 | - |
| CTC | | | - | 38.0 | 61.5 | 40.0 | 7.1 | 7.4 |
| HMM | 0.7 | 0.0 | 0.3 | 66.3 | 31.2 | 71.2 | 6.6 | 7.1 |
| | 0.5 | | 0.0 | 218.0 | 1.0 | 102.1 | 6.8 | 7.2 |
| | | 0.1 | 0.0 | 139.0 | 1.0 | 120.0 | 7.0 | 7.3 |
| | | | 0.1 | 194.6 | 0.8 | 102.1 | 6.8 | 7.2 |

Table 14: Comparison of different blank penalty shifts and posterior and prior scales on our baseline CTC model with SPM 10k, without blank separation and without normed gradient. TSE is w.r.t. the same GMM alignment as in Tables 2 and 3. The reference GMM alignment has 18.0% silence ratio. Posterior scale $\alpha = 1$ always.

| Prior scale $\beta$ | Blank logit shift | TSE [ms] LR | Center | Sil. ratio [%] |
|---|---|---|---|---|
| 0.0 | 0 | 111.5 | 52.9 | 80.8 |
| | -5 | 110.7 | 53.0 | 78.6 |
| | -10 | 93.1 | 47.2 | 59.3 |
| | -15 | 93.7 | 61.1 | 37.3 |
| | -18 | 134.7 | 104.8 | 18.2 |
| 0.0 | 0 | 111.5 | 52.9 | 80.8 |
| 1.0 | | 98.2 | 47.1 | 69.6 |
| 1.5 | | 86.0 | 58.9 | 48.1 |
| 2.0 | | 74.3 | 62.2 | 24.2 |
| 3.0 | | 320.1 | 301.1 | 0.0 |
| 1.0 | 0 | 98.2 | 47.1 | 69.6 |
| | -5 | 73.0 | 46.2 | 44.8 |
| | -10 | 68.2 | 52.0 | 21.8 |
| | -15 | 105.6 | 89.6 | 1.2 |
| | -20 | 110.7 | 94.7 | 0.0 |

**Speed comparison** We benchmark[12] the speed for greedy decoding (either only getting labels, or getting labels with probabilities) and training with a random fixed given alignment which has 90% blank frames, only measuring the final linear transformation from model dimension to vocabulary dimension and the potential log softmax. In case of greedy decoding, we can first calculating the logits for blank, without doing the full linear transformation for the other logits, and then skip this frame when the blank probability is already larger than 50%. The results are given in Table 15. We see quite nice improvements in all cases. For the overall training time, it depends on the encoder, how much percentage of the compute occurs in the encoder and how much in the final transformation and softmax.

---

[12]The code of the benchmark will be published.

Table 15: **Speed** comparison of **blank separation** vs. a full softmax on CTC models. The speedup is calculated as $\frac{\text{full softmax time}}{\text{blank sep. time}}$. Training is with a random fixed given alignment which has 90% blank frames. This is set up in a way that we have a batch size of 10 sequences, a sequence length of 1000, and a vocabulary size of 10000. This is evaluated on a NVIDIA A10 GPU.

| Type of computation | Blank separation | Timings | |
|---|---|---|---|
| | | Absolute [ms] | Speedup [×] |
| Greedy decode only labels | No | 11.4 | - |
| | Yes | 5.2 | 2.2 |
| Greedy decode with probs. | No | 15.8 | - |
| | Yes | 5.3 | 3.0 |
| Framewise training | No | 37.2 | - |
| | Yes | 5.7 | 6.5 |

## A.8 GRADIENT-BASED ALIGNMENTS

**GradScoreExt definition**   Define GradScoreExt as:

$$G' = \log \text{softmax}_{\bar{t}}(G) \in \mathbb{R}^{S \times T} \qquad \text{(softmax over time)} \qquad (44)$$

$$g = \log \frac{1}{S} \sum_s \exp(G')_s \in \mathbb{R}^T \qquad \text{(non-blank score)} \qquad (45)$$

$$l = \text{percentile}(g, \gamma_{\text{percentile}}) \in \mathbb{R} \qquad \text{(flip point)} \qquad (46)$$

$$g' = 2 \cdot l - g \in \mathbb{R} \qquad \text{(blank score)} \qquad (47)$$

$$(G'', g'') = \log \text{softmax}_{S+1}((G', g')) \qquad \text{(softmax over labels incl. blank)} \qquad (48)$$

$$\text{GradScoreExt}(r_t) = \begin{cases} (G'')_{Y_{r_t}, t}, & Y_{r_t} \neq \epsilon, \\ g'', & Y_{r_t} = \epsilon \end{cases} \qquad (49)$$

Here, $\gamma_{\text{percentile}}$ is a hyperparameter (usually $\gamma_{\text{percentile}} = 60\%$).

**Influence of zeroing the blank logits gradient for CTC models**   As a tweak, we slightly modify the gradients of the logits by masking out the gradients of the blank logit. I.e. in the automatic differentiation, we hook after the gradient computation of $\nabla_z L$, and then

$$(\nabla_z L)_\epsilon \leftarrow 0. \qquad (50)$$

See Table 16 for a comparison.

Table 16: Comparing the influence of zeroing out the blank logits gradient. TSE for CTC gradient-based alignments, using $p = 0.1$. SPM10k vocab. The reference GMM alignment has 18.0% silence ratio.

| Align Variant | Mask blank gradient | TSE [ms] | | Sil. ratio [%] |
|---|---|---|---|---|
| | | LR | Center | |
| GradScore | No | 91.0 | 67.4 | 24.8 |
| | Yes | 86.5 | 63.7 | 24.9 |
| GradScoreExt | No | 89.6 | 67.4 | 16.6 |
| | Yes | 84.0 | 62.8 | 15.9 |

**Hybrid AED/CTC**   All our CTC models use an auxiliary AED loss (but without using more data) (Hentschel et al., 2024), thus they can be used as hybrid AED/CTC models (Hori et al., 2017). We can also use the joint probability for the gradient score, using the joint score as defined by Hori et al. (2017):

$$G_{s,t} := \log \left\| \nabla_{x_t} \left( \lambda_{\text{CTC}} \log p_{\text{CTC}}(\bar{a}_s \mid \bar{a}_1^{s-1}, x_1^{T'}) + \lambda_{\text{AED}} \log p_{\text{AED}}(\bar{a}_s \mid \bar{a}_1^{s-1}, x_1^{T'}) \right) \right\|_p \in \mathbb{R}. \quad (51)$$

usually with $\lambda_{\text{CTC}} + \lambda_{\text{AED}} = 1$.

Results are in Table 17. We see that the optimal weighting is reached with $\lambda_{\text{CTC}} = 0, \lambda_{\text{AED}} = 1$, i.e. only the AED is used.

Table 17: TSE for hybrid AED/CTC gradient-based alignments. These are the same CTC models (with joint/aux. AED loss) as in Tables 4 and 5. We use $p = 0.1$ for the grad norm. The CTC model does not use a blank penalty and also no prior here.

| Vocab. | Method | Align Variant | Scale $\lambda_{\text{AED}}$ | Scale $\lambda_{\text{CTC}}$ | TSE [ms] LR | Center | Sil. ratio [%] |
|---|---|---|---|---|---|---|---|
| Reference GMM alignment | | | | | 0 | 0 | 18.0 |
| SPM 10k | - | GradScore | 0.0 | 1.0 | 86.5 | 63.7 | 24.9 |
| | | | 0.1 | 0.9 | 87.1 | 64.3 | 24.9 |
| | | | 0.2 | 0.8 | 86.5 | 63.6 | 25.0 |
| | | | 0.3 | 0.7 | 86.5 | 63.4 | 25.0 |
| | | | 0.4 | 0.6 | 85.0 | 62.5 | 25.1 |
| | | | 0.5 | 0.5 | 82.9 | 61.0 | 25.1 |
| | | | 0.6 | 0.4 | 80.9 | 59.9 | 25.0 |
| | | | 0.7 | 0.3 | 78.7 | 58.7 | 25.0 |
| | | | 0.8 | 0.2 | 74.9 | 56.1 | 24.9 |
| | | | 0.9 | 0.1 | 70.9 | 53.0 | 24.8 |
| | | | 1.0 | 0.0 | 66.6 | 49.5 | 24.6 |
| | | GradScoreExt | 0.0 | 1.0 | 84.0 | 62.8 | 15.9 |
| | | | 0.1 | 0.9 | 84.4 | 63.2 | 15.9 |
| | | | 0.2 | 0.8 | 84.4 | 63.1 | 15.9 |
| | | | 0.3 | 0.7 | 84.1 | 62.8 | 15.8 |
| | | | 0.4 | 0.6 | 82.8 | 61.8 | 15.7 |
| | | | 0.8 | 0.2 | 70.7 | 53.5 | 15.1 |
| | | | 0.9 | 0.1 | 66.9 | 50.8 | 14.8 |
| | | | 1.0 | 0.0 | 63.4 | 48.0 | 14.4 |
| BPE 10k | Blank sep | GradScore | 0.0 | 1.0 | 72.7 | 55.3 | 23.8 |
| | | | 0.7 | 0.3 | 73.4 | 57.8 | 24.5 |
| | | | 0.8 | 0.2 | 73.4 | 57.8 | 24.4 |
| | | | 0.9 | 0.1 | 72.5 | 56.8 | 24.4 |
| | | | 1.0 | 0.0 | 71.6 | 55.9 | 24.4 |
| | | GradScoreExt | 0.0 | 1.0 | 67.3 | 51.1 | 15.2 |
| | | | 0.7 | 0.3 | 64.4 | 50.1 | 14.6 |
| | | | 0.8 | 0.2 | 63.6 | 49.2 | 14.6 |
| | | | 0.9 | 0.1 | 62.9 | 48.4 | 14.6 |
| | | | 1.0 | 0.0 | 61.9 | 47.4 | 14.6 |

A.9 COMPARISON OF ALIGNMENT METHODS

See Figure 5 for a comparison of the presented alignment methods, and Figure 6 for the influence of blank penalty and prior.

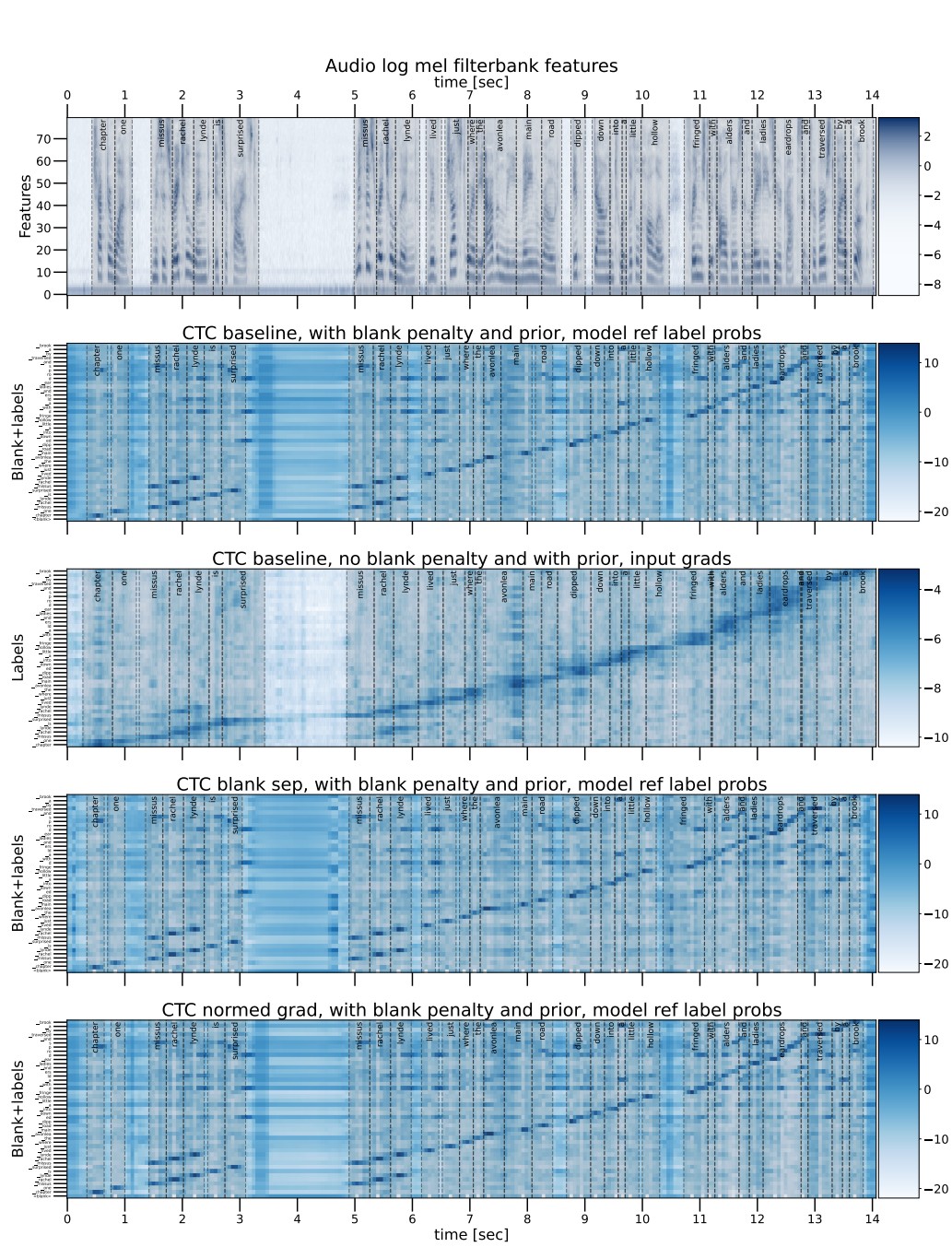

Figure 5: Comparing the different alignment methods. On the top, there are the log mel audio features together with the word boundaries of the reference GMM alignment. Sequence train-clean-100/103-1240-0000.

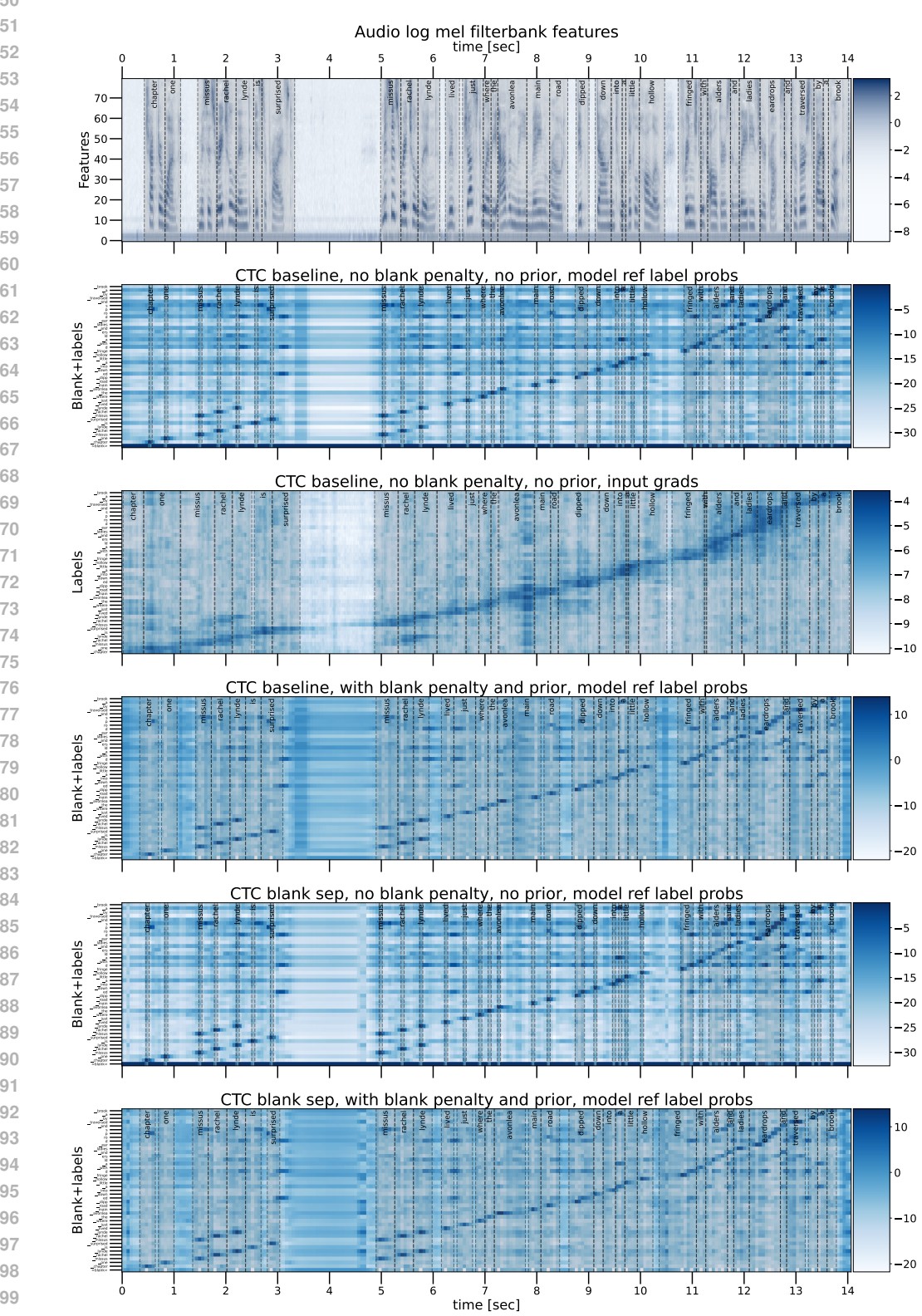

Figure 6: Comparing the different alignment methods, using blank penalty or not, and prior or not. On the top, there are the log mel audio features together with the word boundaries of the reference GMM alignment. Sequence train-clean-100/103-1240-0000.

