# OpenReview forum: "On more accurate alignment modeling methods for automatic speech recognition"
_ICLR.cc/2025/Conference — ICLR 2025 Conference Withdrawn Submission_

### Official Review · Reviewer_ms5i · 2024-10-30

**Soundness:** 3
**Presentation:** 2
**Contribution:** 2
**Rating:** 3
**Confidence:** 5

**Summary:**

The paper proposed several methods to alleviate the peaky behaviour in CTC training. The methods seem to provide more accurate alignment for ASR training with the CTC objective. The authors claimed the contributions from 4 aspects. 1). providing a framework to study alignment behavior based on artificially generated data, and compare various model, noise and training conditions. 2) proposing a new training variant: normalized gradients as an alternative to training with prior. 3) using a novel CTC model variation: Separating the blank label in CTC, as another alternative to counteract class imbalance, leading to improved alignment quality. 4) proposing to us a novel way to get alignments via the gradients of the label.

The storytelling of the proposed methods is not well organized and not that coherent. From my personal understanding, I don't see an obvious gain for some methods and some contributions claimed are not that significant. See comments below for details.

**Strengths:**

The strengths of the paper are:

1. The proposed methods are novel and interesting. We do see more accurate alignment with the method of separating CTC blank.
2. A good connection between the CTC and HMM models.

**Weaknesses:**

The weaknesses of the paper are:

1. The paper is not well organized, making it hard to follow the main contribution and the most significant part of the paper.
2. Some of the claims in the contribution is weak as I observed from the results.

**Questions:**

Here are details for my comments and suggestions to improving the quality of the paper.

1. Clarity:

a. Please provide a definition of TSE. Is it an average number over words or? I expect it to be the lower, the better.

b. What is Fw CE in Table? Please clarify.

c. I don't quite understand the role of synthetic data section. It looks like the experiments with synthetic data are used to obtain insights of training with different prior, posterior and transition scales. Table 1 shows that CTC (alpha=1, beta, and gamma=0) doesn't work for the synthetic data. However, similar ablations have been conducted on Switchboard dataset in Table 1 and of course CTC would work. Another example is using prior is helpful in the synthetic data but not that obvious on Switchboard data. I doubt if Table 1 and the contribution of using synthetic data as a useful tool are really helpful here.

d. It is confusing that the authors using different reference alignments to compute the TSE score, e.g. GMM alignment in Table 2 and CTC forced-alignment in Table 3. This would make the numbers not comparable to each other.

e. The AED model is used when evaluating the effectiveness of the proposed gradient-based alignments, making the entire paper very distracted. First, ablations on different training dynamics on HMM and CTC model without talking about improving alignment quality. Second, proposing grad norm and separating blank on CTC models. Lastly, using gradient-based alignment on CTC and AED. A natural question would be how does grad norm and separating blank CTC work on the hybrid CTC/AED framework since AED model is mentioned and studied in the paper.

2. Performance

a. I am not sure if I understand the metric quality, but in table 4, it looks like blanksep would improve the alignment quality while normed grad would improve the ASR performance as a prior. Are the rows with norm grad and Blanksep separately or incrementally added on top of baseline? If separately, do you have experiments to combine the two? If incrementally, it looks like the normed grad would remove the alignment quality gain brought by blanksep. Can authors give more explanations on this?

b. It is hard to sense the alignment quality improvements. In table 4, the TSE decreases from 111ms to 98ms, which is around 1.5 frames. I am not expecting it as a significant improvement, maybe I am wrong. It would be great if the authors can showcase the reference and generated alignments so that the readers can have a better understanding of the improvements.

c. It is sad to see no big ASR performance gain with the proposed methods. The alignment quality improvement can also be significant though.

---

> ### Author Response · Authors · 2024-11-22
>
> Thanks a lot for the review and the constructive feedback!
>
> Regarding your questions:
>
> > 1a. Please provide a definition of TSE. Is it an average number over words or? I expect it to be the lower, the better.
>
> Yes, it’s the sum of distance between start and end frames for each word, divided by number of words times 2:
>
> $$
> \frac{\sum_w |t_{w,\text{start},\text{ref}}-t_{w,\text{start},\text{model}}|+|t_{w,\text{end},\text{ref}}-t_{w,\text{end},\text{model}}|}{2 \cdot N_{\text{words}}}
> $$
>
> This sum is over some set of sequences. The lower, the better.
>
> We added that also to the paper.
>
>
> > 1b What is Fw CE in Table
>
> For these experiments, we have a ground truth alignment by construction. We calculate the framewise CE w.r.t. the ground truth alignment.
>
> $$
> L_{\textrm{CE}} = - \sum_{t=1}^T \log p(y_t \mid h_t)
> $$
>
> We made this more clear in the newly uploaded version of the paper.
>
>
> > 1c Table 1 shows that CTC (alpha=1, beta, and gamma=0) doesn't work for the synthetic data.
>
> This is for HMM label topology, which is crucial here. For CTC label topology, it works. This is consistent to experiments on Switchboard, where it also does not work with alpha=1, beta, and gamma=0 and HMM label topology, but it works with CTC label topology.
>
> We make more clear in the paper that this is about the HMM label topology, and we added some discussion and experiments on the difference of HMM and CTC label topology in the appendix (see new Table 12).
>
>
> > 1c using prior is helpful in the synthetic data but not that obvious on Switchboard data
>
> The best TSE is also obtained without prior but with transitions on synth data. Regarding WER/LER performance: The variance we get here for the synth data is high, while the LERs are low, so the difference of with and without prior is not significant here.
>
>
> > 1c I don't quite understand the role of synthetic data section. It looks like the experiments with synthetic data are used to obtain insights of training with different prior, posterior and transition scales.
>
> The motivation is to design a synthetic framework where experiments can be done and insights can be gained to much better understand the training behavior and alignment behavior on a wide range of settings, both covering realistic settings but then also covering more extreme settings, to get a better understanding whether some method is sound and stable in principle or not.
>
> The effect of different scales is just one example, but there is a lot more. We summarized some of the main findings, but we put some more results into the appendix.
>
> Most of the experiments we show are consistent to the findings on real data. And also it gave us better understanding on the alignment behavior, when a training criterion would each a good alignment. We found it very interesting that even with the trivial input-output mapping in the synthetic data, many training criteria and models will not produce good alignments. But there is a lot of further potential in this framework to study certain aspects in more detail, and we also see the need to extend the simulated data distribution for certain cases to make it more realistic.
>
> As we plan to release this framework together with the paper, I think this is an important contribution to everyone who wants to study the training and alignment behavior for CTC, HMM or similar kind of models.
>
>
> > 1d It is confusing that the authors using different reference alignments to compute the TSE score, e.g. GMM alignment in Table 2 and CTC forced-alignment in Table 3.
>
> This was an unclear formulation in the Table 3 caption. The reference alignment in the whole paper is always the same GMM alignment. Table 3 compares this GMM ref alignment to a CTC forced alignment. We reformulated the table caption, that this is about the CTC model, and uploaded a new PDF.
>
> > 1e. A natural question would be how does grad norm and separating blank CTC work on the hybrid CTC/AED framework since AED model is mentioned and studied in the paper.
>
> Yes, this is a good idea. This can be done. We added this to the paper in appendix. We tested different scales. Interestingly, the best TSE is obtained when only the AED is used.
>
> (I will make a separate post with further answers.)

---

> ### Author Response · Authors · 2024-11-22
>
> > 2.a  Are the rows with norm grad and Blanksep separately or incrementally added on top of baseline?
>
> Separate.
>
> > 2.a If separately, do you have experiments to combine the two
>
> Yes, we did that. We added that to the table as well. There is no clear improvement over each.
>
> > 2.b. TSE decreases from 111ms to 98ms, which is around 1.5 frames
>
> It’s correct that this is only a minor difference in TSE. But the TSE alone does not tell the full story  about the alignment quality. The amount of silence frames also gives you an important indicator on the alignment, and for this specific example, the difference is huge. This can have significant effects on downstream tasks.
>
> We uploaded a new version where we added an example alignment plot which show the difference, where the TSE diff is small but the amount of silence is huge.
>
> > 2c It is sad to see no big ASR performance gain with the proposed methods. The alignment quality improvement can also be significant though.
>
> Yes that is true…
>
>
> > The storytelling of the proposed methods is not well organized and not that coherent.
>
> Do you maybe have suggestions on how to improve that? I understand that there are several separate methods proposed here, and this maybe makes it difficult to keep a good overview. But we thought they are still all related to each other in many ways.

---

### Official Review · Reviewer_pvoH · 2024-11-03

**Soundness:** 3
**Presentation:** 2
**Contribution:** 2
**Rating:** 6
**Confidence:** 4

**Summary:**

This submission focuses on the important goal of improving the alignment model for ASR, primarily for the CTC framework, but with consideration of the AED framework too. A synthetic data experimental framework is adopted as part of the investigation, in addition to evaluations performed on the well-known public domain datasets, Switchboard and LibriSpeech. New alignment modeling techniques are proposed: (1) a gradient normalization method aimed at class balancing, (2) a blank-factored CTC model, and (3) a gradient-based approach to obtaining the gradients for both CTC and AED. The results suggest that the blank-factored CTC model yields slightly more accurate time alignments, and slightly better WERs too; and that the gradient-based method for obtaining alignments improves alignment quality for larger vocabularies.

**Strengths:**

Improving alignment quality is a highly relevant practical topic in the field of ASR, and this work provides value in evaluating a number of reasonable variants of current modeling methods. The evaluation includes use of controllable synthetic data framework, as well as well-known public benchmarks. Some small but significant improves in alignment quality and WER are presented.

**Weaknesses:**

The presentation and writing seem a bit rough, and in particular, the motivation for some of the methods proposed is not expressed very clearly -- though the reader can speculate and fill in the blanks. The motivations stated in the Abstract and Introduction seem a bit vague. I don't disagree with most of the overall points made, but I think the reasoning could be made tighter and clearer. I think the 3 new model methods proposed can reasonably be expected to improve alignment quality, but it's not really detailed exactly why that is so... E.g. the notion of "imbalance". Just because e.g. the blank symbol or silence token occur very very frequently, why is that necessarily bad for a "naive", non-blank-factorized model's alignments? I can speculate, but ideally there would be a clearer argument than provided by the paper. Similarly for the proposed gradient based renormalization, the authors write: "Instead of the prior (which is e.g. estimated on the average of p(y | h)), now we use the prior estimated on the average of the soft alignment υ." Sounds reasonable, but what is the theoretical or practical advantage compared to using a standard prior normalization, as previously proposed in the literature? Same question for the gradient-based obtaining of the gradients. For AED, the advantage is clear, since there is no explicit notion of alignment for AED. But what is the theoretical or practical advantage for CTC?

Regarding the Synthetic Data setup, in Section 6.1, I think it would be helpful to summarize the essential properties of the setup before going into the details. What are the specific dimensions that the authors wish to control, that the synthetic data setup provides?

**Questions:**

In addition to the questions I mention above, some specific comments/questions on the text:

In the Abstract:

"Hidden Markov models (HMMs) can be seen as a generalization of CTC": usually we think of the CTC model as existing within the broader HMM framework, not the other way around.

"Label units such as subword units and its vocabulary size...": it is not clear what "its" refers to here.

L037: "The classic speech recognition models such as Gaussian mixture hidden Markov model (GM-HMM) and later the hybrid neural network (NN)-HMMs (Bourlard & Morgan, 1993) rely on frame-wise cross-entropy training on a single best alignment path": (1) GMM-HMMs do not use cross-entropy training; (2) and both GMM-HMMs and Hybrid ASR DNN/HMMs have often been trained with dynamic programming to sum over all alignment paths. Embedding dynamic programming into the optimization process has been a standard tool for HMM-based ASR for decades, see e.g. Rabiner & Juang, "Fundamentals of Speech Recognition", 1993. (I suppose we can disagree on what is the "most classic" approach, but it seems to me the statement is too strong).

L045: "HMMs can be trained with the sum over all alignments as well, and differ from CTC only by label topology." To me this is a type mismatch: CTC exist within the HMM family, so "HMMs" actually includes a multitude of topologies, including CTC. It's like saying, "Animals can live in many different environments, and differ from cats only in what they eat."

L071: "word-error-rate" --> "word error rate"

L121: "averaged of the posterior" --> "average of the posterior"

L167: "This is very related to the training criterion with a prior: Instead of the prior (which is e.g. estimated on the average of p(y | h)), now we use the prior estimated on the average of the soft alignment υ": I agree, but what is the advantage?

L203: "In this case, the classes in pA are much more balanced compared to the classes in pY , as blank is usually the most imbalanced class": blank is the more frequent class, certainly, but what does it mean for a class to be imbalanced...? "Imbalanced" is a negative value judgment, but the authors don't flesh out why e.g. a very frequent class poses a problem to alignment modeling -- though the reader might agree with the statement, it should be motivated with a specific theoretical or practical intuition.

L373: "7.1.2 PHONEME-BASED MODELS" Clarify early on that the results in this section will use Switchboard and LibriSpeech, not the synthetic data?

L399. "Here, we use less number of epochs" --> "Here, we use fewer epochs" or "Here, we use a smaller number of epochs"

Re: Switchboard and LibriSpeech in 7.1.2: though the authors provide citations for these in the Appendix, it seems the citations should appear in the section? This section is a slightly odd mix of self-contained and not self-contained, in the sense that the Switchboard results are in Table 2 of the main body, the LibriSpeech results discussed are in Table 10 of the Appendix.

L531: "framerate" --> "frame rate"

---

> ### Author Response · Authors · 2024-11-23
>
> Thank you for your review and the feedback!
>
> We reformulated parts of the abstract, introduction and conclusion to make the motivation more clear: We want to use a good performing model to generate good quality alignments.
>
> > Just because e.g. the blank symbol or silence token occur very very frequently, why is that necessarily bad for a "naive", non-blank-factorized model's alignments?
>
> (And your comment on L203)
>
> There is a whole research field which demonstrates the problems with class imbalance and proposes solutions to it. We cited some, e.g.:
>
> - Kaidi Cao, Colin Wei, Adrien Gaidon, Nikos Arechiga, and Tengyu Ma. Learning imbalanced datasets with label-distribution-aware margin loss, 2019. URL https://arxiv.org/abs/1906.07413
> - Tsung-Yi Lin, Priya Goyal, Ross Girshick, Kaiming He, and Piotr Dolla ́r. Focal loss for dense object detection, 2018. URL https://arxiv.org/abs/1708.02002.
>
> We also added now some more, e.g.:
>
> - Justin M Johnson and Taghi M Khoshgoftaar. Survey on deep learning with class imbalance. Journal of big data, 6(1): 1–54, 2019.
> - Wuxing Chen, Kaixiang Yang, Zhiwen Yu, Yifan Shi, and CL Chen. A survey on imbalanced
> learning: latest research, applications and future directions. *Artificial Intelligence Review*, 57(6): 1–51, 2024.
>
> Deep learning algorithms can fare poorly when the training dataset suffers from heavy class-imbalance, e.g. that it does not properly converge or poor performance. With careful hyper parameter tuning, we mostly made it work anyway (the community trains ASR models successfully since a long time), but we think that this also negatively influences alignment behavior, training dynamics, training robustness, and such methods can improve this.
>
> We extended the discussion on related work about this in Section A.1.
>
> > "Instead of the prior (which is e.g. estimated on the average of p(y | h)), now we use the prior estimated on the average of the soft alignment υ." Sounds reasonable, but what is the theoretical or practical advantage compared to using a standard prior normalization, as previously proposed in the literature?
>
> (And also your comment on L167)
>
> At the end of section 3, we give some theoretical explanation:
>
> Consider also the case of very clean synthetic data together with a simple single-layer feed-forward neural network (FFNN) where we can initialize $W = \operatorname{identity}$ and $b=0$. This initialization will provide a perfect alignment for this synthetic task. It will stay perfect as long as $b$ stays uniform. Now, $\nabla_b L_{\textrm{CTC}}$ is not uniform, thus the model will not keep good alignment behavior. But $\nabla_b L_{\textrm{NormedGradCTC}}$ is uniform by construction.
>
> We also expanded this further in the paper: When using CTC with prior, $\nabla_b L$ would also not be uniform, i.e. $L_{\textrm{NormedGradCTC}}$ is really the best possible loss you can have here.
>
> Regarding the practical advantage, this is what we try to show with the experiments. And we do improve slightly over a very strong baseline. Although the improvements are quite small. But this is probably because the baseline was already well tuned. On a not-so-well-tuned baseline, the method might give more improvements.
>
> > But what is the theoretical or practical advantage for CTC [for the gradient-based alignment]?
>
> There are multiple reasons:
>
> The encoder is so powerful that it can do many strange things, like shifting around the signal (e.g. often with streaming models), even reversing the direction (https://arxiv.org/abs/2410.00680). Even in those cases, the gradient-based alignment will always be reasonable, as this is the gradient w.r.t. the input signal. Thus it should be much more robust and more generic.
>
> The gradient-based alignment is calculated on the input feature frame rate (often 100 Hz), while the model output frame rate is often downsampled (e.g. 25 Hz), thus you can get a higher resolution for the alignment.
>
> The gradient-based alignment method is so generic that it also can be used for many different kinds of applications. The presented application here is just one example to demonstrate this.
>
> We also expanded this motivation in the paper.
>
> (More in next comment.)

---

> ### Author Response · Authors · 2024-11-23
>
> > Regarding the Synthetic Data setup, in Section 6.1, I think it would be helpful to summarize the essential properties of the setup before going into the details. What are the specific dimensions that the authors wish to control, that the synthetic data setup provides?
>
> You are right, this is a better structure. We now moved the detailed description of the data sampling to the appendix, and instead provide a summarization of what we control. Specifically:
>
> - The ground truth alignment. We construct the input features accordingly.
> - Noise in the input features.
> - The vocabulary and labels, and statistics on how many words per sequence.
> - Statistics about how much silence there is and the duration of labels. This indirectly simu- lates different framerates of the input features.
>
> Then we support a variety of model types (GMM, CTC, hybrid HMM; various neural encoders; different prior model variants; transition probabilities), CTC and HMM label topology, and different training criteria.
>
> > "Hidden Markov models (HMMs) can be seen as a generalization of CTC": usually we think of the CTC model as existing within the broader HMM framework, not the other way around.
>
> English is not our native language, but what we wrote is exactly the same as what you write? HMM can be seen as a generalization of CTC, i.e. HMM is more general, i.e. CTC can be seen as a special case of the broader HMM framework.
>
> But we reformulated this whole part in the introduction now to make this hopefully more clear.
>
>
> > "Label units such as subword units and its vocabulary size...": it is not clear what "its" refers to here.
>
> “Its” refers to the subword units / label units. For e.g. BPE or SPM, you can control the vocabulary size, i.e. the amount of labels. I’m not sure exactly how to reformulate that in a way that it does not sound strange? Or maybe "vocabulary size" is the misleading terminology here?
>
> > L037: "The classic speech recognition models such as Gaussian mixture hidden Markov model (GM-HMM) and later the hybrid neural network (NN)-HMMs (Bourlard & Morgan, 1993) rely on frame-wise cross-entropy training on a single best alignment path": (1) GMM-HMMs do not use cross-entropy training; (2) and both GMM-HMMs and Hybrid ASR DNN/HMMs have often been trained with dynamic programming to sum over all alignment paths. Embedding dynamic programming into the optimization process has been a standard tool for HMM-based ASR for decades, see e.g. Rabiner & Juang, "Fundamentals of Speech Recognition", 1993. (I suppose we can disagree on what is the "most classic" approach, but it seems to me the statement is too strong).
>
> Yes, our formulation was wrong. Definitely GMMs do not use CE training. And yes, hybrid NN/HMMs also have been trained with sum over all paths. We agree with all what you say. Our argument here was more about what is more standard, but this is of course debatable. We reformulated this whole part in the introduction now.
>
> > L045: "HMMs can be trained with the sum over all alignments as well, and differ from CTC only by label topology." To me this is a type mismatch: CTC exist within the HMM family, so "HMMs" actually includes a multitude of topologies, including CTC.
>
> Yes, also this formulation was misleading. We basically wanted to say the same as what you also explained. We now reformulated huge parts of the introduction to make this hopefully more clear.
>
> > L373: "7.1.2 PHONEME-BASED MODELS" Clarify early on that the results in this section will use Switchboard and LibriSpeech, not the synthetic data?
>
> Yes, we made this more clear now.

---

> > ### Comment · Reviewer_pvoH · 2024-12-03
> >
> > Thanks for addressing my comments. I think the new version is definitely improved, but I will keep my original score.
> >
> > > English is not our native language, but what we wrote is exactly the same as what you write? HMM can be seen as a generalization of CTC, i.e. HMM is more general, i.e. CTC can be seen as a special case of the broader HMM framework.
> >
> > I think we fundamentally agree -- but since HMMs were invented (much) earlier than CTC, it would be more natural to say "CTC can be seen as a specific HMM topology", rather than "HMM can be seen as a generalization of CTC".
> >
> > In that light, it still seems a bit odd, throughout the paper, to refer to a particular model as the "HMM label topology", and contrast that conceptually and experimentally with CTC. We agree that HMMs are a superset of CTC, so contrasting "HMM" and "CTC" is like contrasting "animals" with "elephants".

---

### Official Review · Reviewer_N5Du · 2024-11-04

**Soundness:** 2
**Presentation:** 3
**Contribution:** 2
**Rating:** 3
**Confidence:** 3

**Summary:**

CTC training commonly leads to peaky behavior where the model predicts blank in most frames and the labels are focused mostly on single frames. Therefore, CTC is suboptimal in obtaining accurate word boundaries. Gaussian mixture hidden Markov models are typically used to obtain reliable and accurate segment and word boundaries.

This paper proposes modifications to the CTC training criterion and training procedure to improve alignment quality. The paper proposes normalized gradients as an alternative to training with label priors to improve CTC training. The paper also proposes to separate the blank label in CTC loss calculation to counteract class imbalance.

**Strengths:**

The paper is well-written and systematically builds the improvements to the CTC to reduce the peakiness in CTC alignments.

**Weaknesses:**

My main critique of the work is that the proposed method doesn't seem to be all that effective at improving the alignment quality. For example,

Section 7.2, "NORMALIZED GRADIENT", L460 "Unexpectedly, there does not seem to be any improvement in terms of alignment quality (TSE). Also, in terms of convergence rate, there was no difference"
Section 7.3, "BLANK SEPARATION IN CTC" while the left-right boundaries improve by separating the blank, there is still a significant gap between the CTC alignments and GMM reference alignment. There seems to be no improvement for the word center positions. There is no improvement in the convergence rate and the reduction in WER is very small (<0.2).

Overall the results seem significantly worse than GMHMM alignments.

**Questions:**

Does the baseline CTC (e.g. in Table 4,5 ) use frame-level priors during training? If not, why not? Why are previous works (Zeyer et al., 2021; Chen et al., 2023; Huang et al., 2024) not included in the result tables?

---

> ### Author Response · Authors · 2024-11-23
>
> Thank you for your review and the feedback!
>
> I think we can clarify some misunderstanding on the comparison of the GMM alignment vs the other alignments in terms of TSE:
>
> The reference GMM alignment here has the best TSE (TSE 0) by definition, because we calculate the TSE with respect to the reference GMM alignment. So, based on these TSE numbers, it does not make sense to say that the reference GMM alignment has better TSE numbers than the other alignments, because it obviously has that by definition.
>
> You could argue that the use of a GMM alignment as reference for the TSE calculation itself is problematic. We do this because we do not have another good reference alignment on Switchboard or Librispeech. Thus, this measure of TSE will never tell us whether we really get better than the GMM alignment.
>
> Note, in the literature, there have been other approaches to measure the alignment quality:
>
> - Train another model on top of this and measure its WER. But this can be problematic, as we know that sometimes a worse alignment can result in better training due to regularization effects.
> - Evaluate on a different dataset where we have a human-annotated reference alignment, e.g. TIMIT or Buckeye.
>
> Also note, we agree with you and we also still think that the GMM alignment is better. Here we don’t expect to get better alignments than the GMM. Instead, we want to keep a good performing model (in terms of WER) and improve its alignment quality to get a bit closer to the GMM alignment. This is different from related work in the literature on obtaining better quality alignments, where the models often perform bad in terms of WER (just like the GMM also produces bad WERs).
>
> We rewrote parts of the introduction to make this hopefully more clear.
>
> We also added a figure where we compare the different alignments (reference GMM alignment, CTC forced alignment, gradient-based alignment), which should make the issues of CTC more clear, and how we improve on that.
>
> > Does the baseline CTC (e.g. in Table 4,5 ) use frame-level priors during training? If not, why not? Why are previous works (Zeyer et al., 2021; Chen et al., 2023; Huang et al., 2024) not included in the result tables?
>
> For our setup with conformer-based models the CTC phoneme based systems had worse WER and TSE using prior, however, this was an initial investigation. We are running further experiments.
>
> The previous works are not directly comparable in the presented TSE measure here (they evaluate on different corpora, or use other metrics; and in any case not the same GMM reference alignment).
>
> Note also, e.g. Huang et al. 2024 trains a very tiny model (5M params TDNN), which assumably has very bad WER performance. Our motivation here is to keep a good performing ASR model. When a bigger model is trained with prior, more problems occur on training stability and performance (both WER and alignment quality).
>
> > My main critique of the work is that the proposed method doesn't seem to be all that effective at improving the alignment quality.
>
> We agree that the overall improvement is not large.
>
> Most interesting is maybe the gradient-based alignment method, which is very generic, and also works fine when the encoder does weird things like shifting around the signal.
>
> Also, the synthetic framework can serve as a valuable tool for researchers interested on the training dynamics.

---

### Official Review · Reviewer_oKY6 · 2024-11-06

**Soundness:** 3
**Presentation:** 3
**Contribution:** 2
**Rating:** 6
**Confidence:** 2

**Summary:**

The paper tackles the problem of CTC models for ASR and the poor quality of the word-level alignment due to the use of a "blank" token, no explicit silence label and "peaky" output labels. To help ameliorate the issue, the authors introduce an artificial task to study alignment behavior of various systems under tightly controlled conditions, ie. the amount of silence and distribution of word start and end times are set precisely to yield known distributions.

The contributions of the paper include synthesizing a task for diagnostic purposes, a renormalized gradient training regime for CTC to counteract the well known class-imbalance with the blank token, reformulating the CTC output into a hierarchical softmax, then blank/non-blank, as alternate mitigation  for blank class-imbalance, and lastly computing the alignment from gradients. The experiments demonstrate the usefulness of the mixing the transition, prior and posterior probabilities and the alignment are generaly robust to these values as long as they are not too extreme (e.g. \alpha=1.0, \beta=\gamma=0.); also the hierchical softmax "separated blank", normalized gradient and alignment via gradient were all found to be beneficial in generated better alignments with CTC models.

**Strengths:**

Well written and motivated with a strong set of experiment and analyses on well-controlled synthetic task and realistic test sets.
Contributions are good for a widely used class of model (CTC) and address an important aspect of the model ie alignment quaility for transcription.

**Weaknesses:**

The paper focuses on a narrow subject for the ICLR community, namely transcription alignment quality for CTC modeling. Its hard to make a constructive comment on how to make this work more broadly appealing to the general ICLR audience.

**Questions:**

In Table 1, the penultimate line where \alpha=0.5 is better than \alpha=1.0 is this because the posterior scores are too sharp? Do you have any intuition on why this scaling performs better?

In Table 2 of this paper,  the CTC models have worse TSE 89.5ms on the same dataset in https://arxiv.org/pdf/2407.11641 (Table 1, best 38ms). What are the differences in the training/modeling that explain the difference in TSE?

In Section 7.3 the paper reports on Blank Separation and no improvement in convergence rate. Another reported benefit is computational; while it may implementation dependent, do you have any results that demonstrate efficiency gained from the hierarchical softmax?

---

> ### Author Response · Authors · 2024-11-23
>
> Thank you for your review and feedback!
>
> > The paper focuses on a narrow subject for the ICLR community, namely transcription alignment quality for CTC modeling. Its hard to make a constructive comment on how to make this work more broadly appealing to the general ICLR audience.
>
> It is true that we focus here on the alignment quality of ASR models.
>
> The presented gradient-based alignment method is however generic, and can be applied to other applications as well. E.g. it can be used to generate alignments for machine translation.
>
> > In Table 1, the penultimate line where \alpha=0.5 is better than \alpha=1.0 is this because the posterior scores are too sharp? Do you have any intuition on why this scaling performs better?
>
> Yes, the posterior scores become very sharp, and this helps. Also note that it mostly helps in the case when we have a combination of models like am together with transition and/or prior.
>
> > In Table 2 of this paper, the CTC models have worse TSE 89.5ms on the same dataset in https://arxiv.org/pdf/2407.11641 (Table 1, best 38ms). What are the differences in the training/modeling that explain the difference in TSE?
>
> We had a typo in the column title of Table 2, this is actually SWB300h and not LBS 960h.
>
> Moreover, the mentioned paper is using BLSTM encoder and not Conformer.
>
> > In Section 7.3 the paper reports on Blank Separation and no improvement in convergence rate. Another reported benefit is computational; while it may implementation dependent, do you have any results that demonstrate efficiency gained from the hierarchical softmax?
>
> We added some speed comparison for blank separated models, for greedy decoding and framewise CE training, demonstrating huge speedups (new Table 15). E.g. the framewise CE training is speed up by factor 6, and also greedy decoding is 2-3 times faster. This speedup is only considering the final linear transformation and potential softmax, though. Everything which comes before that is shared, thus there is no difference. It thus depends on how much the final part takes of the total compute. This depends on the type and size of encoder, and also the vocabulary size.

---

> > ### Comment · Reviewer_oKY6 · 2024-11-25
> >
> > Thank you for your replies and clarifications especially Table 15. The huge speed-ups are indeed a nice win.
> >
> > However, I have not observed any substantial changes either to either raise or lower my score, so I keep my score as is. Thanks again.

---

### Author Response · Authors · 2024-11-23
**New paper version**

Note that we uploaded a new version of the paper. We did several changes:

- All questions and suggestions have been addressed in the current uploaded draft. (If you think we missed some aspect which was not addressed, please tell us.)
- Alignment plots comparing CTC forced alignment vs gradient-based alignment (new Fig 5 and Fig 6)
- Automata for CTC and HMM label topologies (new Fig 3 and Fig 4)
- More experimental results
    - Comparison between HMM and CTC for synth data (new Table 12)
    - Hybrid AED/CTC result for gradient-based alignment (new Table 17)
    - TSE results using different prior scales and blank penalty during alignment (new Table 14)
    - Speed comparison for blank separated models, for greedy decoding and framewise CE training, demonstrating huge speedups (new Table 15)
    - Updated many TSE numbers, as they improved when using prior everywhere (Table 3, 4, 5)
    - Added an experiment where we used blank separation and normalized gradient together (but it does not give further improvements) (Table 4)
- Mathematical formulation
    - Time-stamp-error (TSE)
    - Framewise cross-entropy (Fw CE)
- Expanded related work section
- Reformulated abstract, introduction and conclusion, to emphasize more one of the core motivations: We want to keep a good performing model here (CTC with good WER) and getting good alignments from this model. In the literature, often the model with good alignment quality has suboptimal WER, and vice-versa. This is different here.

It turns out, when using priors during alignment generation, all the TSEs improve, both for the CTC forced alignment, and also to a lesser degree to the gradient alignments. The blank penalty further improves the CTC forced alignment. Now, with the prior, there is no improvement in alignment quality anymore for the blank separation and the normalized gradients. These new results certainly lower the significance of blank separation and normalized gradients.

The gradient-based alignment still gives us some small improvement over the forced alignment. We think this is still interesting in itself, that it works so well at all, and shows the potential of the gradient-based alignment. It also works even in the case that the model shifts around the alignment a lot, i.e. when the forced alignment quality would be bad (which is not so much the case for the presented model here). We also note that this gradient-based alignment can be used for any kind of model, and also for other tasks, such as alignments in machine translation. Also, here it seems the alignment quality is correlated to the WER, in contrast to many other methods.

We also think there is value in the presented synthetic framework to study alignment behavior and training dynamics.

---

### Note · Authors · 2024-12-03

**Comment:**

We thank once again all the reviewers for their valuable feedback! We withdraw this submission for multiple reasons:

* We think we can further improve the submission to resolve most of the raised concerns.
* We noticed some inconsistencies in the gradient computation in some of our experiments, which needs further investigations, and might invalidate some of the experimental results using separated blank and normed gradient with CTC on Librispeech. (Those are anyway the results where we got less improvement than what we expected.)

For this, we need some more time.

**Withdrawal Confirmation:**

I have read and agree with the venue's withdrawal policy on behalf of myself and my co-authors.